# Adaptation to host cell environment during experimental evolution of Zika virus

Vincent Grass [1], Emilie Hardy[1,7], Kassian Kobert[2,7], Soheil Rastgou Talemi [3,7], Elodie Décembre [1], Coralie Guy[1], Peter V. Markov[2], Alain Kohl [4], Mathilde Paris[5], Anja Böckmann[6], Sara Muñoz-González [1], Lee Sherry [1], Thomas Höfer [3], Bastien Boussau [2✉] & Marlène Dreux [1✉]

Zika virus (ZIKV) infection can cause important developmental and neurological defects in Humans. Type I/III interferon responses control ZIKV infection and pathological processes, yet the virus has evolved various mechanisms to defeat these host responses. Here, we established a pipeline to delineate at high-resolution the genetic evolution of ZIKV in a controlled host cell environment. We uncovered that serially passaged ZIKV acquired increased infectivity and simultaneously developed a resistance to TLR3-induced restriction. We built a mathematical model that suggests that the increased infectivity is due to a reduced time-lag between infection and viral replication. We found that this adaptation is cell-type specific, suggesting that different cell environments may drive viral evolution along different routes. Deep-sequencing of ZIKV populations pinpointed mutations whose increased frequencies temporally coincide with the acquisition of the adapted phenotype. We functionally validated S455L, a substitution in ZIKV envelope (E) protein, recapitulating the adapted phenotype. Its positioning on the E structure suggests a putative function in protein refolding/stability. Taken together, our results uncovered ZIKV adaptations to the cellular environment leading to accelerated replication onset coupled with resistance to TLR3-induced antiviral response. Our work provides insights into Zika virus adaptation to host cells and immune escape mechanisms.

[1] CIRI, Inserm, U1111, Université Claude Bernard Lyon 1, CNRS, UMR5308, École Normale Supérieure de Lyon, Univ Lyon, Lyon 69007, France. [2] Laboratoire de Biométrie et Biologie Évolutive (LBBE), UMR CNRS 5558, Université Claude Bernard Lyon 1, Lyon 69622, France. [3] Theoretical Systems Biology, German Cancer Research Center, Deutsches Krebsforschungszentrum (DKFZ) Heidelberg, Heidelberg 69120, Germany. [4] MRC-University of Glasgow Centre for Virus Research, Glasgow G61 1QH, UK. [5] Institut de Génomique Fonctionnelle de Lyon (IGFL), École Normale Supérieure de Lyon, Lyon 69007, France. [6] Institut de Biologie et Chimie des Protéines, MMSB, Labex Ecofect, UMR 5086 CNRS, Université de Lyon, Lyon 69007, France. [7] These authors contributed equally: Emilie Hardy, Kassian Kobert, Soheil Rastgou Talemi. ✉email: Bastien.Boussau@univ-lyon1.fr; marlene.dreux@ens-lyon.fr

Zika virus (ZIKV; *Flaviviridae*) is a mosquito-borne human pathogen related to other globally relevant flaviviruses, including dengue, yellow fever, West Nile, Japanese encephalitis, and tick-borne encephalitis viruses. As is typical for flaviviruses, ZIKV has a 10.8 kb RNA genome of positive polarity, encoding a polyprotein composed of 3 structural proteins (C, prM and E) and 7 nonstructural (NS) proteins. The NS proteins are involved in the steps of RNA synthesis and assembly of viral particles. Several NS proteins of flaviviruses interfere with host antiviral responses, either by inhibition of the innate sensors or downstream signaling pathways[1–4].

For decades, ZIKV infections were either unrecognized or occurred only sporadically and were associated with mild symptoms. However, ZIKV was detected in Brazil in 2015 and spread rapidly, reaching infection rates exceeding 50%[5]. During the Brazilian ZIKV outbreak, congenital infections led to fetal demise, microcephaly, and other developmental abnormalities (now grouped as Congenital Zika Syndrome), e.g., visual and hearing impairment, skeletal deformities, and possibly Guillain–Barré syndrome in adults[5–9]. Severe symptoms, including neural development defects and fetal demise, are linked to host antiviral responses by type I and III interferons (IFN-I/III), which are also central for ZIKV control and in utero transmission[10–15]. All cells possess signaling pathways designed to trigger the production of IFN-I/III and IFN-stimulated genes (ISGs) upon viral infection. Their effects are potent and wide-ranging: direct inhibition of the viral life cycle at multiple steps and jumpstart of the adaptive immune response. These antiviral responses are induced by the recognition of specific viral motifs by host sensors, such as Toll-like receptors (TLR) or RIG-I-like receptors (RLR), that mobilize cascade signaling. As for other flaviviruses, TLR3-induced signaling reduces ZIKV replication[16]. Nonetheless, like virtually all human pathogenic viruses, ZIKV has evolved the ability to modulate and counteract the IFN-I/III signaling and other host responses[1–4], likely through interactions with host proteins. The mutation rate of ZIKV is expected to be around $10^{-4}$ to $10^{-5}$ mutation per site per replication in accordance with other flaviviruses, since the catalytic site of the NS5 polymerase is well-conserved among flaviviruses[17]. This mutation rate ensures high genetic diversity within hosts, and adaptability of viral populations. Adaptive mutations that improve the fitness of the virus can do so by improving the viral machinery, optimizing the interactions with proviral host factors, or inhibiting antiviral factors. Given the limited size of the viral genome, trade-offs between these three strategies are expected, especially since they play out in both human and mosquito hosts.

Previous studies on ZIKV evolution focused on the ability of the virus to maintain robust replication in the context of alternate human/mosquito hosts[18]. Nonetheless, clinical studies have demonstrated the ability of arboviruses, including ZIKV, to replicate and last for several months in a subset of infected patients (e.g., detection of viral genome in plasma, urine, and semen)[19–24], underlining the need to better understand the possible outcome(s) of viral evolution in the human host.

Molecular tracking of arbovirus evolution in the host is greatly complicated by the error rate of the polymerases used for sequencing, which makes it difficult to distinguish between low-frequency genetic variants that have appeared during evolution and sequencing errors. To overcome this challenge, methods for accurate identification of ultra-rare and low-frequency genetic variants have recently been developed. Especially, the CirSeq method, which involves generating tandem repeats from circularized RNA templates, has proved successful to analyze the viral genetic diversity of poliovirus[25] and of DENV[26], as well as the landscape of transcription errors in eukaryotic cells[27].

Here, we adapted this methodology to the arbovirus ZIKV, to reveal how it can adapt to the human host cell environment and to investigate the genetic interactions involved. We conducted an in-depth analysis of the evolution of viral populations (i.e., population of genomes isolated from infected cell supernatants) through serial passages of a Brazilian ZIKV strain in human cell cultures. This led us to uncover a phenotypic change linked to higher viral spread, *via* increased specific infectivity, which is associated to viral resistance to TLR3-induced antiviral responses. Bioinformatic analyses showed that specific ZIKV variants increased in frequency in temporal association with this phenotypic adaptation, and the corresponding mutations were functionally validated.

## Results

**Acquisition of increased specific infectivity during experimental evolution.** We performed experimental evolution in a human hepatocyte cell line (Huh7.5.1. cells) as it has been previously described to be highly efficient in ZIKV production. This cell type suffers little cytopathogenic effect of ZIKV infection and is non-responsive to viral products, as it is deficient for different antiviral response sensors[28–31]. A Brazilian patient isolate (ZIKV Pernambuco PE243; KX197192) was chosen, which means that the starting inoculum is more diverse than if it had been derived from a ZIKV molecular clone. We sequenced the starting inoculum, and found that the major variant in our viral preparation had on average a frequency of 99.5% (2.5% quantile: 99.2%, 97.5% quantile: 99.9%), and thus there was a standing diversity of 0.5% on average at each position. All positions matched the published consensus sequence except for position 1786, which had major variant T instead of C, probably from the initial amplification in Vero cells of the clinical isolate. Experimental evolution was performed by serial passaging of ZIKV: at each passage (up to 18 passages), viral populations harvested at 3 days post-infection were used to infect naïve cells (Fig. 1a, schema on the left side), using MOI 0.1 for the first infection by parental virus and MOI 0.01 for subsequent viral passaging, because of the limited viral yield from early passages (see 'Method' section). Quantification of infectious virus produced over the course of the experimental evolution showed that viral production increased during serial viral passaging and reached a plateau by passage 7-to-9 (Supplementary Fig. 1a). Increased viral productions were observed for all 3 independent runs of experimental evolution and within a similar timeframe (Supplementary Fig. 1a). Likewise, quantifications of intracellular and extracellular viral RNA levels confirmed the augmentation of ZIKV replication over time (Supplementary Fig. 1b). Passaging of the virus from runs #2 and #3 was stopped when viral production reached the level at which run #1 plateaued.

To address how passaged viral populations adapted to human host cells, we first studied their ability to initiate infection as compared to the parental virus by quantifying the specific infectivity, defined as the probability for one physical virion to initiate infection, i.e., the ratio of the number of infectious viruses relative to the total amount of vRNA (as described in Methods section). We demonstrated an increase of the specific infectivity of the viral populations harvested over the course of all independent runs of experimental evolution (Fig. 1a). The maximum level of specific infectivity was observed by passage 7-to-8 in the independent runs of experimental evolution followed by a plateau. The trend of increased viral production (Supplementary Fig. 1a) appeared to be simultaneous and in proportion with the augmentation of specific infectivity (Fig. 1a), suggesting that the ability of ZIKV to adapt in this defined host environment likely occurred *via* an increased capacity to initiate infection.

**Increased specific infectivity of passaged virus enables resistance to TLR3-induced antiviral response.** Next, we tested the capacity of these passaged viral populations to propagate when

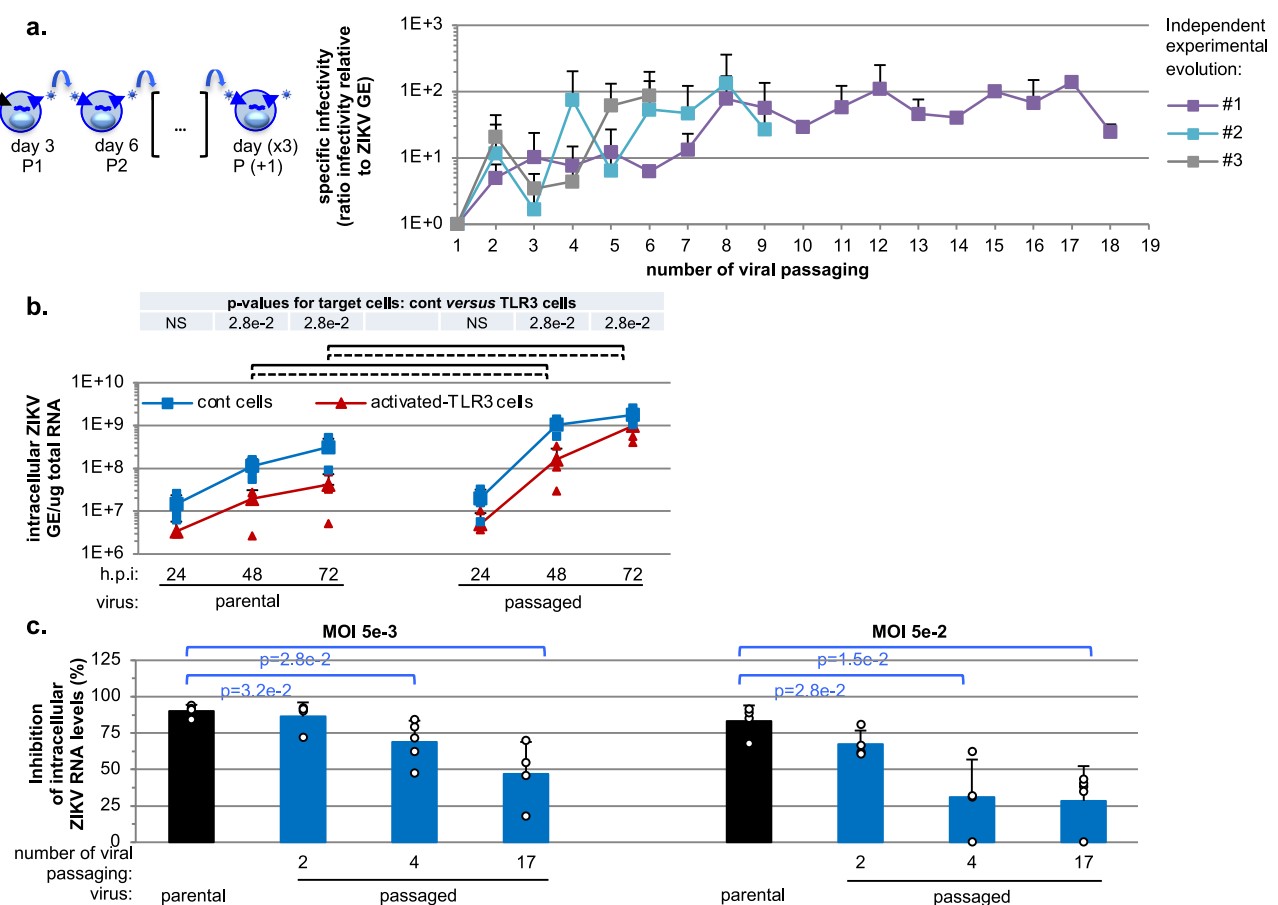

**Fig. 1 Increased specific infectivity and resistance to TLR3-mediated inhibition of serially passaged viral populations. a** As shown on the schematic representation of the experimental procedure for serial passaging of ZIKV viral populations: every 3 days, viral supernatants were harvested, infectivity levels determined and, accordingly, used to infect naïve cells. Quantification of the specific infectivity (i.e., defined as the probability for one physical virion to initiate infection) of the viral populations harvested over serially passaging in the course of 3 independent runs of experimental evolution (referred to as #1, #2, #3 and stopped at passages 18, 7, and 6 respectively). Results are expressed as the ratio of the extracellular infectivity levels relative to extracellular ZIKV RNA levels and relative to passage 1 for each independent passaging in control cells set to 1; 2-to-3 independent determinations by RT-qPCR and infectious titrations for each evolution experiment; mean ± SD. **b** Quantification of the intracellular ZIKV genome levels in kinetic analysis of serially passaged viral populations obtained at passage 17 of the viral passaging versus parental virus assessed in activated-TLR3 cells (dotted lines) as compared to control cells (solid lines); at the indicated times post-infection at MOI 0.05; 4 independent experiments; mean ± SD. The statistical comparison of intracellular ZIKV GE levels for the same viral population at the same time point post-infection between activated-TLR3 and control cells is performed using a one-way ANOVA on ranks (Kruskal−Wallis test), as in ref. [31]. When the test was considered significant (p-values ≤ 0.05), we used the Tukey Kramer (Nemenyi) pairwise test as post hoc test for pairwise comparisons of mean rank sums to determine which contrasts between individual experimental condition pairs were significant. The p-values from the Tukey Kramer (Nemenyi) pairwise tests are indicated in the table at the top of the graphs, when $p < 0.05$. The intracellular ZIKV GE levels at a same time post-infection between parental and passaged virus similarly determined as significantly different ($p < 0.05$) are indicated by brackets: dotted lines for the comparison of levels in activated-TLR3 cells and solid lines for the comparison of levels in control cells. **c** Inhibition by TLR3-signaling of replication of viral populations harvested at the indicated times of the serial passaging (i.e., passages 2, 4, and 17). Results represent the percentage of inhibition in activated-TLR3 cells relative to control cells of intracellular ZIKV genome levels determined at 3 days post-infection at MOI 5e$^{-3}$ (left panel) and 5e$^{-2}$ (right panel); 4 independent experiments; mean ± SD; significant differences ($p < 0.05$) are indicated by the brackets at the top of the graphs.

submitted to the host antiviral responses. Huh7.5.1 cells are known to be deficient for TLR3-induced signaling[29], enabling specific induction of the antiviral response by complementation *via* ectopic expression of wild-type (WT) TLR3 (Supplementary Fig. 1c). Treatment of cells expressing WT TLR3 by poly(I:C), a mimetic of the intermediate double-stranded RNA produced during viral replication, led to a robust ISG upregulation at mRNA and protein levels, by using ISG15 and MxA as representative ISGs. ISG15 is considered to be part of the early response, being preferentially induced directly by transcriptional factors such as IRF3, downstream of sensor-induced signaling, and by IFN-I/III receptor-induced signaling. MxA expression belongs to the late response regulated mostly through IFN-I/III

receptor-induced signaling[32] (Supplementary Fig. 1d, e). In contrast, the parental Huh7.5.1 cells, without ectopic WT TLR3, did not respond to poly(I:C) (Supplementary Fig. 1d, e). This demonstrated that WT TLR3 expression renders our cell model responsive to the TLR3 agonist poly(I:C). Specificity of the TLR3-induced ISG response was also confirmed by the absence of ISG upregulation upon poly(I:C) treatment of cells expressing TLR3 with a deletion of the Toll/interleukin-1 receptor (TIR) domain of the cytosolic tail (ΔTIR-TLR3), necessary for recruiting the downstream TIR domain-containing adapter inducing IFNβ (TRIF)[33] (Supplementary Fig. 1c, d).

By using this set up, we showed that poly(I:C)-induced TLR3 signaling greatly decreases replication of the parental virus,

but not that of passaged viral populations harvested at passage 17 (Fig. 1b, c). Importantly, viral RNA levels measured at 72 h post-infection were similar in the TLR3-activated cells versus the non-activated cells for the passaged virus (Fig. 1b, right curves). We quantified the inhibition of viral replication by activated-TLR3 signaling for viral populations harvested at different time points (Fig. 1c). In keeping with the results shown in Fig. 1b, TLR3-induced antiviral response inhibited the parental virus replication by up to 90% at 72 h post-infection. The resistance to TLR3-mediated inhibition was already observed for the viral populations harvested at passage 4 and at the different MOIs applied (Fig. 1c), thus likely independent of the infection of one cell by infectious units containing several virions[34].

The resistance to TLR3-induced antiviral response can result from a faster onset of infection by the passaged viral populations. In such a scenario, ongoing replication before the establishment of a robust antiviral response in host cells would out-compete the latter. In support of this hypothesis, the replication rate of passaged viral populations was faster compared to the parental virus in the same infection set-up (MOI 0.05) and in the absence of TLR3-induced response (Fig. 1b, solid lines and comparing the slope from the time-point 24 h to 48 h post-infection). Similar observations were made for viral populations harvested at distinct late time points of the experimental evolution, including at passage 12 (Supplementary Fig. 1f). These observations suggest that the passaged viral populations could outcompete host antiviral response by an earlier replication onset post-inoculation.

**Increased viral replication depends on the targeted cell type but is independent of the ISG response.** Faster replication, and thus a faster expression of viral proteins compared to host anti-viral effectors may be cell-type specific. To test this hypothesis, we assessed the infection efficiency of passaged virus across different cell environments. First, to assess the infection speed by the passaged viral population compared to the parental virus, we quantified the size of infectious foci formed in a given timeframe, as reflecting the propagation speed *via* rounds of infection. Foci formed upon infection of Huh7.5.1 cells by the long-term passaged viral population were significantly larger than those formed by parental virus (Fig. 2a, b, left panels). This confirmed the increased infection speed of the viral populations in the cell type used for experimental evolution of ZIKV. The opposite was observed in the simian Vero cells (Fig. 2a, b, right panels). To further analyze this cell type-specific phenotype, we performed a kinetic analysis of viral replication. The replication rate of passaged viral populations significantly increased (approx. 10-fold) compared to parental virus in Huh7.5.1 cells (Fig. 2c, upper panel), with similar ZIKV RNA levels observed early after infection, likely reflecting the viral input. In contrast, the replication rates of the adapted viral population did not increase compared to parental virus in Vero cells (Fig. 2d, upper panel). The kinetic analysis of the ISG response (representative MxA and ISG56 mRNAs) demonstrated a difference between the cell types (Fig. 2c, d, lower panels and Supplementary Fig. 2b, c). An early ISG response was triggered upon infection of Huh7.5.1 cells by parental virus, yet only observed with MOI ≥ 0.1, and vanished for the viral populations harvested later in the course of the experimental evolution (Fig. 2c and Supplementary Fig. 1f, g). In Vero cells, infection by either viral population led to similar ISG responses (Fig. 2d and Supplementary Fig. 2c). This differential profile of ISG response between the two cell types might result from the absence in Vero cells of a response to an activating signal contained in the supernatants harvested at early time-points of the experimental evolution, or alternatively from qualitatively and/or quantitatively different cell entry pathways in these cell types.

To discriminate these possibilities, we broadened the phenotypic analysis to alternative human cell types. Similar to Huh7.5.1 cells, the replication rate of adapted viral populations was clearly increased in HEK-293 cells compared to the parental virus (Fig. 2e, upper panel). Unlike Huh7.5.1 cells, neither parental virus nor adapted viral populations induced an ISG response in HEK-293 cells (Fig. 2e and Supplementary Fig. 2d), indicating that higher replication rate of adapted viral populations versus parental virus can occur independently of an ISG response. The results also demonstrated that the increased propagation rate of the adapted viral populations is not restricted to a unique cell type. To further confirm that the increased infection rate by the adapted viral population is independent from the early ISG response, we compared the viral replication kinetics in both human U6A cells deficient for STAT2 (i.e., an ISG transcription regulator of the signaling pathway induced by IFN-I and III), and U6A cells complemented for STAT2 expression, referred to as STAT2-U6A cells (Fig. 2f, upper panels). STAT2 expression potentiates the ISG response upon ZIKV infection with an accelerated response observed at 48 h post-infection in STAT2-U6A cells as opposed to the poor response of the corresponding U6A cells (Fig. 2f and Supplementary Fig. 2e) and consistently, decreased viral replication in STAT2-U6A cells compared to the corresponding U6A cells (Fig. 2f, comparing the upper panels). In both the U6A cells and STAT2-U6A cells, the levels of both viral replication and the ISG response were similar for the adapted versus the parental viral populations (Fig. 2f). Taken together, our results obtained in cell types failing to respond to ZIKV infection by ISG upregulation, such as HEK-293 cells, or STAT2-U6A cells competent for ISG induction, suggested that the increased infection rate of the adapted viral populations is independent of the extent of ISG upregulation.

We also tested the rate of viral replication in macrophages differentiated from monocytes isolated from healthy human blood donors (Fig. 2g and Supplementary Fig. 2a), representing an in vitro cell model closely related to the cell type targeted in vivo by ZIKV[35–38]. Interestingly, we observed that the propagation of the passaged viral populations was significantly abrogated in this cellular model, compared to the parental virus (Fig. 2g). Conversely, ISG induction was readily detected in response to the parental virus, but not for the passaged viral population (Fig. 2g and Supplementary Fig. 2f).

These comparative analyses with different cell types suggested that the viral adaptations leading to higher infection rate are independent of the ISG response, and that the ability of passaged viral populations to initiate infection most likely involves viral-host interactions and entry pathways that depend on the cellular environment.

**Genetic diversity and evolution of the viral populations determined by deep-sequencing analysis.** To delve into the mechanism underlying viral adaptation, we analyzed the genomic diversity of the viral populations harvested in the course of evolutionary experiments. Sanger sequencing and next-generation deep-sequencing methods generally provide consensus sequences. The error rate of next-generation sequencing can limit the detection of low-frequency variants because rare variants and sequencing errors may have similar frequencies. Yet, low-frequency variants can nevertheless be very important functionally. To bypass this limitation we adapted the "CirSeq" method that reduces next-generation sequencing errors[25] to sequence viral variants at various time points of our three runs of experimental evolution. We used total RNA derived from approx. 5x10E6 to 5x10E7 secreted infectious particles per

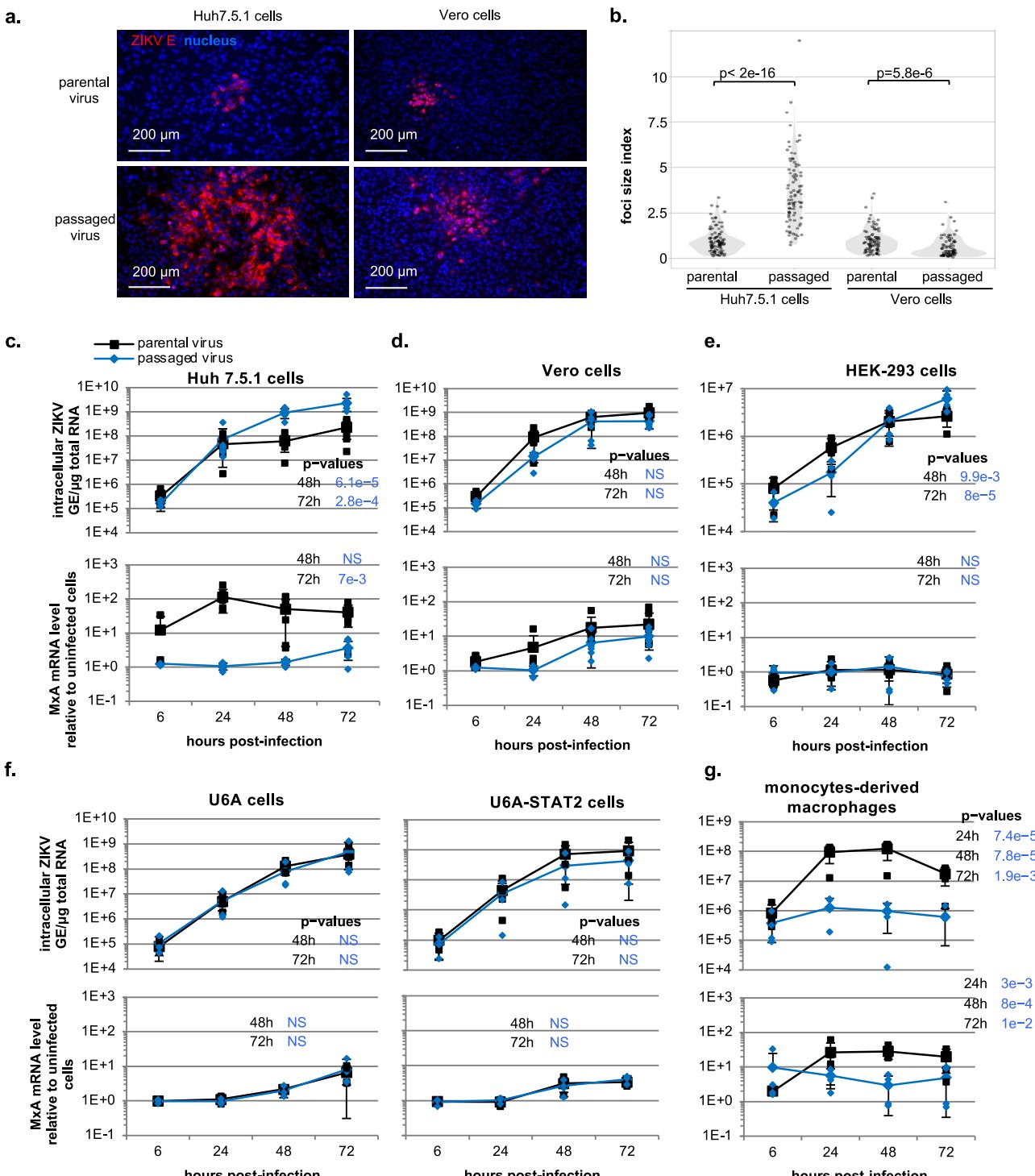

**Fig. 2 Increased viral replication depends on the targeted cell type but is independent of the ISG response. a** Representative imaging of infectious foci at 48 h post-infection by parental virus versus serially passaged virus (i.e., passage 17) in Huh7.5.1 cells (left panels) and Vero cells (right panels) by immunostaining of E envelope proteins (red); nucleus stained by Hoechst (blue); scale-bar indicated. **b** Violin plot representation of the focus size in Huh7.5.1 cells and Vero cells at 48 h post-infection by parental virus versus serially passaged virus harvested at passage 17 of the viral passaging. The size index for each infectious focus is displayed by an individual dot; 5 independent measurements; p-values from pairwise comparisons using Wilcoxon rank sum test are shown. **c–g** Kinetic quantification in Huh7.5.1 (**c**), Vero cells (**d**), HEK-293 cells (**e**), U6A cells and STAT2 expressing U6A cells (**f**) and macrophages derived from monocytes (**g**) post-infection by parental versus serially passaged viral population obtained at passage 17 of the viral passaging. Results present the levels of intracellular ZIKV GE (upper panels) and MxA (lower panels) mRNA levels relative to the levels in non-infected cells, at the indicated time post-infection at MOI 0.1; 3-to-7 independent experiments; mean ± SD. The p-values of the statistical analysis of the kinetics are determined using a mixed linear model, which takes into account the levels at the time-points before those indicated in the table on the right side of the graphs. The indicated p-values are for the comparison of passaged viral population versus parental virus, when $p < 0.05$.

sample. Bioinformatic analyses revealed that CirSeq was successful, with tandem repeat occurrence ranging from 59% to 94% (mean 84%) per read, and repeat sizes ranging from 33 to 97 bases (mean size 50 to 60 bases). In addition, coverage quantification revealed that our methodology using 200-PE runs with HiSeq 2500XL led to a read number/depth as high as 2x10E5 (Fig. 3a and Supplementary Fig. 3a). Minimal coverage, as defined as read numbers below 1000, was detected at only 55 positions out of all the analyzed libraries and the mean coverage per position across all analyzed experimental conditions ranged from approximatively 10E4 to 10E5 (Supplementary Fig. 3b), thus allowing an in-depth analysis of viral populations. Moreover, the profiles of viral genome coverage across all analyzed samples were similar (Fig. 3a and Supplementary Fig. 3a). These results thus validate the reproducibility of our experiments and imply that composition and/or secondary structure of certain viral genome segments had a limited impact on sequencing depth. Next, we analyzed the mutation frequencies over the course of the three independent runs of experimental evolution (Fig. 3b, c and Supplementary Fig. 3c). We validated that the consensus sequence obtained by deep-sequencing of the parental inoculum used for our experimental evolutions perfectly matches the clinical isolate reference sequence (GenBank ID KX197192). To identify mutations whose frequency significantly varied through time, for each run, we computed position-wise standard deviations of the frequency of the most frequent variant. As shown in Supplementary Fig. 4a, the standard deviations are distributed in two categories: (i) a high density of very low-frequency variants, likely corresponding to polymorphisms due to the error-prone feature of the NS5 polymerase and/or generated during library preparation, despite the CirSeq protocol and (ii) low density of polymorphisms showing higher standard deviations, corresponding to mutations that have reached a high frequency during at least one time-point of one experiment. Selecting variants with standard deviation above a threshold of 0.1 or 0.02 showed that 4 or 7 variants, respectively, had high standard deviations in two or three runs (Supplementary Fig. 4b, c). We provide a representation of variant frequencies at passage 6 that highlights the similarity of the frequency profiles between independent experiments (Fig. 3c).

### Identification of candidate adaptive mutations in ZIKV E and NS4B proteins.

We further analyzed the mutations that passed the standard deviation thresholds, considering them as candidate adaptive mutations (Fig. 3d–f and Supplementary Fig. 4b, c). Among the 4 variants that passed the 0.1 threshold, 3 positions do so in a number $(n) = 3$ of experiments (positions 1786, 2341, 7173); 1 additional position does so in $n = 2$ (position 2194; Fig. 3d, e). Two additional positions are found when the threshold is set to 0.02 for $n = 3$ (positions 5663, 10,007; Fig. 3f). To know if some of these variants were already present at the beginning of the experimental evolution, we sequenced the parental viral population. We compared the frequency of our variants of interest to the frequency of all other variants in the library, and computed percentiles. The minor variant frequencies were 12.5% for C1786T, 0.06% for C2194T, 0.3% for C2341T, 3.3% for T5663C, 0.3% for T7173C, 2.0% for C10007T. This places variants C1786T, T5663C, and C10007T among the 1% most frequent variants, C2341T between the 33 and 34% most frequent variants, T7173C between the 44 and 45% most frequent variants, and C2194T between the 97 and 98% most frequent variants (i.e., between the 1 and 2% rarest variants). Variants C1786T, T5663C, and C10007T were thus present in the initial inoculum at the highest frequency. The non-synonymous C1786T mutation was detected within the course of viral adaptation with an increasing frequency over time. This leads to a substitution at position 270 of E envelope protein from V to A. Since A270 is present in the clinical isolate database sequence (ZIKV PE243, KX197192)[39] V270 is likely resulting from initial viral amplification of the clinical isolate in Vero cells, and A270 is actually a reversion when passaged in Huh7.5.1 cells. Variant C2194T was particularly rare, and variants C2341T and T7173C had unremarkable frequencies at the start of the experiment, in the middle of the distribution of variant frequencies. Among the variants, C2341T, T7173C, C1786T, and C2194T are non-synonymous mutations within the coding part of the ZIKV polyprotein and are thus most likely to have phenotypic effects on the virus. The frequencies of mutations C1786T and C2194T do not vary in a manner correlated with the increased specific infectivity of the evolved virus and thus are unlikely to be linked to this phenotype. Mutations C2341T and T7173C may however be linked to the evolved phenotype.

The frequency of the C2341T mutation increased from passage 2-to-4 (Fig. 3d), thus preceding both the augmented specific infectivity and the acquisition of resistance to TLR3-induced antiviral response (Fig. 1a and Supplementary Fig. 1a, b). This corresponds to a substitution of amino acid S-to-L, at position 455 of the E envelope protein. The T7173C mutation also became the majority variant by passages 4-to-6 in the three independent experiments (Fig. 3d). This corresponds to a Y-to-H substitution at position 87 in NS4B.

We performed similar runs of viral evolution experiment upon induction of the TLR3 antiviral response, using Huh7.5.1 cells expressing WT TLR3 stimulated by the agonist poly(I:C) (Supplementary Fig. 5a, b). We sequenced the resulting viral populations in the 3 independent runs. Interestingly, we observed an increase of the frequency of the C2341T and T7173C mutations upon ZIKV passages in this experimental setting (Supplementary Fig. 5c). Together, our bioinformatic analyses identified two point mutations in E and NS4B proteins that might be involved in the phenotype of viral adaptation observed during experimental evolution, both in the absence and in the presence of the TLR3 antiviral response.

### The S455L mutation in the E viral protein recapitulates both the increased specific infectivity and the resistance to TLR3 antiviral response.

To functionally validate the two candidate adaptive mutations, the S455L substitution in the E envelope protein and the Y87H substitution in NS4B were introduced alone, or in combination, into recombinant ZIKV. First, we generated a ZIKV molecular clone (referred to as 'ref no mut') corresponding to the clinical isolate used for experimental evolution by introducing R99G and Y122H substitution in NS1 in the previously reported molecular clone BeH819015[40]. These NS1 mutations had no impact on viral spread and specific infectivity (Supplementary Fig. 6a–c).

The efficiency of transfection of the different mutants was comparable to that of the reference, as determined by ZIKV intracellular levels at 6 h post-transfection (Supplementary Fig. 6g). While the Y87H substitution in NS4B showed no significant difference as compared to the reference clone (ref no mut) (Supplementary Fig. 6d–f), the S455L substitution in the E protein significantly increased the rate of viral replication when present alone or in combination with Y87H in NS4B (Fig. 4a). The S455L substitution in E also recapitulated the enhanced specific infectivity observed for viral populations evolved in the course of the evolution experiments (compare Figs. 4b to 1a).

We examined the speed of viral propagation *via* new rounds of infection by measuring the size of infectious foci formed in a limited timeframe (as in Fig. 2a, b). We observed a clear increase

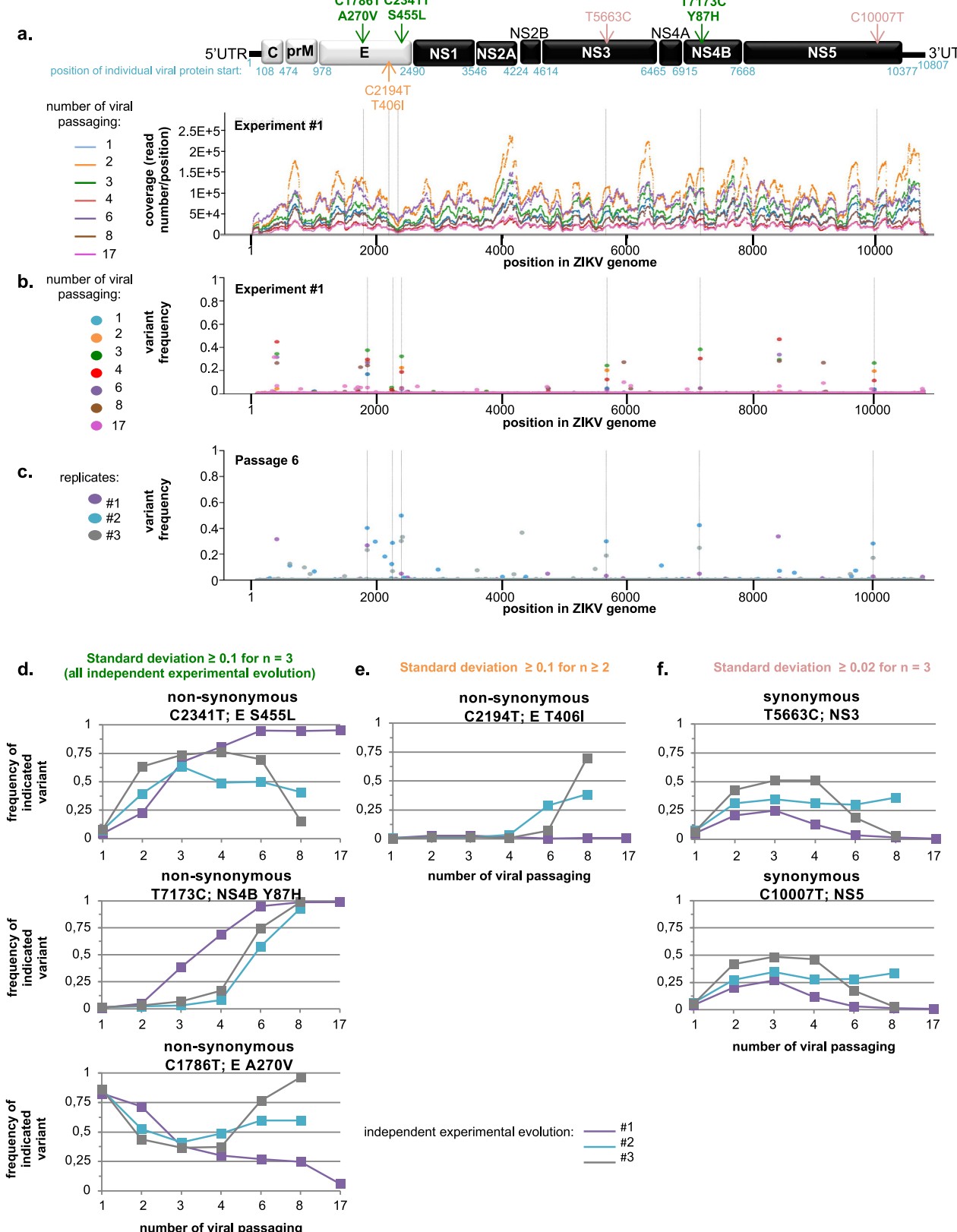

upon infection with the E S455L mutant as compared to the reference in Huh7.5.1 cells (Fig. 4c and Supplementary Fig. 6h, left panels). In contrast, in Vero cells, the size of infectious foci formed upon infection by the same E S455L mutant was reduced as compared to the reference (Fig. 4c and Supplementary Fig. 6h, right panels). This is consistent with the observations for the viral populations adapted in the course of experimental evolution described earlier (Fig. 2a, b). We demonstrated that, similar to the adapted viral populations obtained in the experimental evolution, the E S455L mutant resists inhibition by TLR3-induced antiviral response (Fig. 4d, right curves), as opposed to the reference clone (Fig. 4d, left curves).

**Fig. 3 Bioinformatic analysis of the genetic evolution of viral populations obtained by next-generation sequencing. a** Coverage of the next-generation sequencing analysis along the ZIKV genome sequence of viral populations harvested at the indicated time points of the serial passaging of one representative independent run of the evolution experiment. Results are expressed as number of reads per position; schematic representation of ZIKV genome at the top. **b, c** Time-course quantification of the frequency of the non-parental variants at each position along ZIKV genome in the viral populations harvested in at least one representative independent run of evolution experiment (**b**) and in the viral populations harvested at passage 6 in three independent runs of evolution experiments (**c**). In both cases, the frequency of the second most frequent variant is shown. Dotted lines indicate the positions in the viral genome with high standard deviations in several runs of experimental evolution, as defined in Supplementary Fig. 4c. **d–f** Time-course quantification of the frequency of variants determined by next-generation sequencing. The variants were selected when the standard deviations of their frequencies were: ≥0.1 for all the three independent runs of experimental evolution ($n = 3$; n referred to one replicate of one condition at given time of harvest) (**d**); ≥ 0.1 for a minimum of 2 samples (**e**), and ≥ 0.02 for a minimum of three samples (**f**), with thresholds defined according to the density of variants relative to their frequency for the pool of all analyzed samples, as presented in Supplementary Fig. 4. The variants are indicated as nucleotide position (e.g., C2340T), the corresponding viral protein (e.g., E) and amino acid change for non-synonymous substitution (e.g., S455L); as also shown on the schematic representation of ZIKV genome organization (**a**).

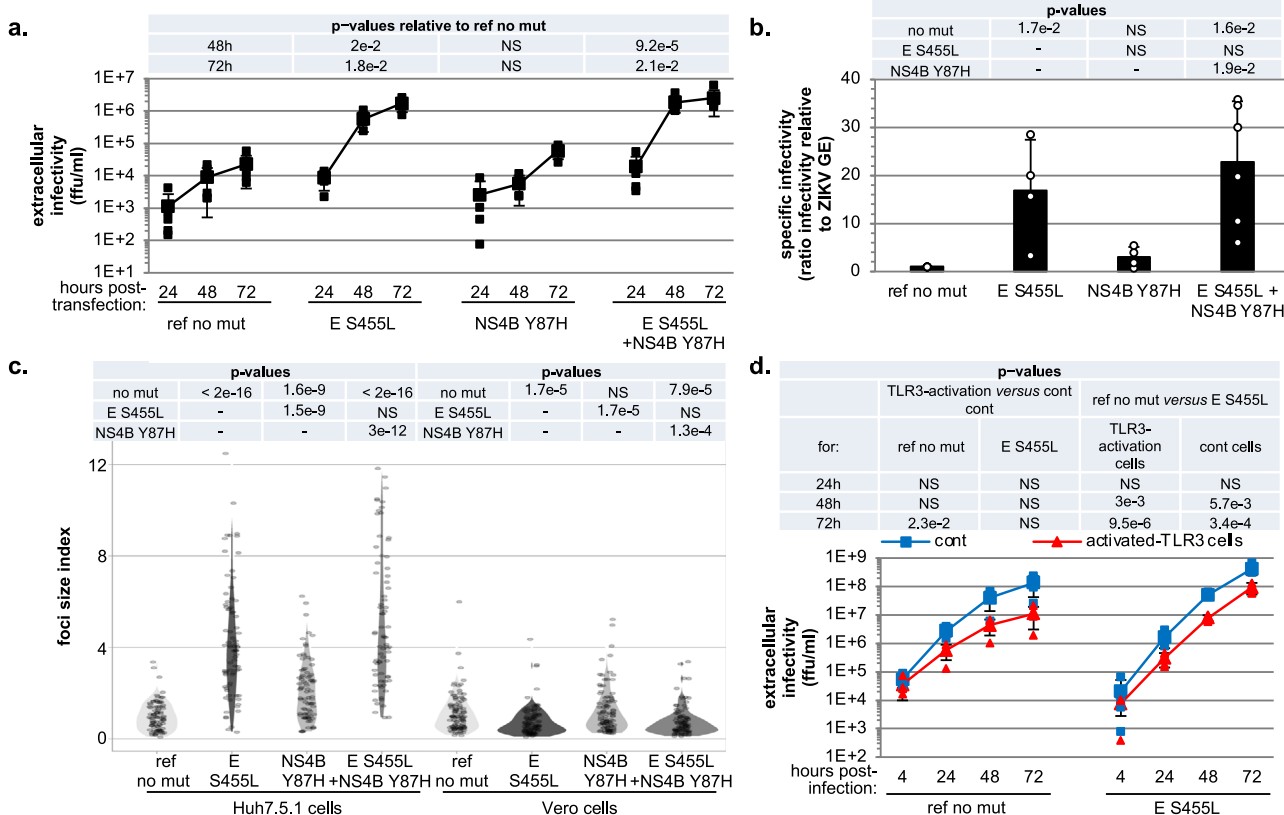

**Fig. 4 Introduction of the selected non-synonymous mutations into a ZIKV molecular clone.** ZIKV genome bearing the selected mutations (i.e., single S455L mutation in E, Y87H in NS4B and combined E S455L and NS4B Y87H mutations), and as a reference ZIKV genome without the mutation (i.e., ref no mut), were transfected in Huh7.5.1 cells. **a** Time-course quantification of infectious viral production at the indicated times post-transfection. Results of 4-to-6 independent experiments; mean ± SD; p-values as indicated in the table above the graph and relative to the reference (ref no mut). **b** Quantification of the specific infectivity (as in Fig. 1a) in the viral supernatants harvested at 72 h for ZIKV genome mutants or not. Results are the mean ± SD relative to the reference virus set to 1 for each independent experiment; 4-to-6 independent experiments. The p-values indicated in the table correspond to one-by-one comparisons of the condition/mutant displayed below in the graph with conditions/mutants indicated on the left side of the table. **c** Analysis of the focus size index of the indicated mutated or reference ZIKV determined in Huh7.5.1 and Vero cells, as indicated, for supernatants harvested at 72 h post-transfection. The quantifications are displayed by violin plots, determined as in Fig. 2b. Results of 4 independent experiments; p-values as indicated in the table above the graph. **d** Time-course analysis of the replication of the E S455L mutant versus reference virus (ref no mut) assessed in activated-TLR3 Huh7.5.1 cells (red lines) as compared to control cells (blue lines). Quantification of the intracellular ZIKV genome levels at the indicated times post-infection at MOI 0.005; 4 independent experiments; mean ± SD. The statistical comparison of intracellular ZIKV GE levels for the same target cells and between mutant and reference virus is indicated in the table at the top of the graphs with indicated p-values and NS: $p > 0.05$.

Given that the E S455L mutation causes improved viral replication both in the presence and absence of TLR3-induced signaling, we asked whether an earlier replication could provide a single mechanism explaining both observations. An earlier replication could be due to two connected processes: either an increased rate of infection (rate hypothesis), or a reduced delay between infection and viral replication (delay hypothesis). We modeled both hypotheses mathematically and compared the models and experimental data. The models share a similar backbone (Fig. 5a and Supplementary Fig. 8a) and describe the infection of susceptible cells, which, after a time delay, enter a phase of productive viral replication. Produced infectious

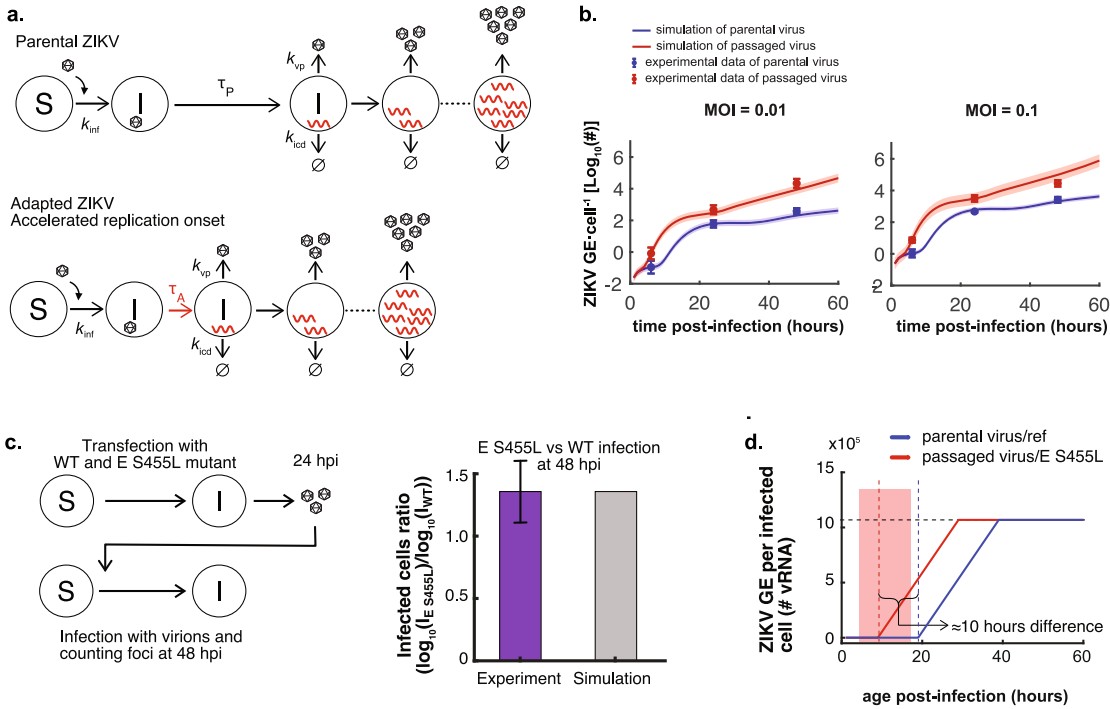

**Fig. 5 Multiscale model of ZIKV infection and replication, with differences in the delay between infection to replication. a** Schematic representation of the model. ZIKV infects susceptible cells with rate $k_{inf}$. Productive viral replication begins with a time delay $\tau_P$ for the parental virus and $\tau_A$ for the evolved or mutated viruses. Virus replication within infected cells is modeled as a piecewise linear function approximating virus replication (Supplementary note 1, mathematical model). Infected cells produce virions with rate $k_{vp}$, and die with rate $k_{icd}$. **b** Model simulation with optimized parameters versus experimental measurements of the number of ZIKV genomes per cell upon infection with different doses of parental and adapted ZIKV. **c** Model simulation versus experimental measurements of the ratio of the infected cells upon infection with wild type and the E S455L mutant ($I_{E\ S455L}/I_{WT}$). Left panel, schematic representation of the procedure for the multiscale model in accordance with the experimental method used in the transfection experiments including analyses of the extracellular infectivity. **d** Simulation of the ZIKV replication inside the infected cells for parental/ref and adapted/E S455L mutant. The shaded region is the 95% CI for the estimated $\tau_A$ value.

virions can then infect other susceptible cells. Individual cells in this multiscale model will have their own time course of ZIKV replication, depending on the time of infection (Supplementary note 1, mathematical model). We determined the kinetic parameters of the models (including infection rate, delay to productive replication, and parameters defining the intracellular RNA level) by fitting the models to two sets of experimental data (Fig. 4, Supplementary note 1, and Supplementary Fig. 6). To test the delay hypothesis, the effects of both evolution and introduction of the E S455L point mutation were modeled by allowing the delay to productive replication to be different from parental and reference strains, respectively, keeping all other parameters identical. To test the rate hypothesis, we allowed the infection rate to be different from parental and reference strains (Supplementary Fig. 8). Both models captured the data well (Fig. 5b, c and Supplementary Fig. 8b, c). Focusing on the delay model, the parameter values were well constrained (Supplementary Fig. 7). Further, the delay to productive replication was shortened from around 20 h for the controls to 10 h or less (best fit 10 h and upper 95% confidence bound 17 h; Supplementary Fig. 7) for the mutated strains (Fig. 5d). Results were similar whether fitted on either data from parental versus evolved strains or data from reference versus mutated strains. Our models thus are consistent with the E S455L substitution either reducing the time-lag between infection and viral replication or increasing the infection rate (or possibly both). Both phenomena appear sufficient to cause increased viral spread, and probably the resistance to TLR3-induced antiviral response.

The position 455 is located at the C-terminus of the stem region, next to the transmembrane domain of the E protein[41–43]. In the structure of E (5ire.pdb reference)[43], the position 455 is located near the membrane surface, in an environment of nearly exclusively polar and hydrophobic residues (Fig. 6a, b). It is located at the end of an amphipathic helix which spans residues 436-454. Its side chain hydroxyl does not engage in interactions with the transmembrane helix of the M protein facing it, but is likely involved in lipid headgroup interactions. A mutation from the polar Ser to the very hydrophobic Leu in this position thus has the potential to markedly change the membrane anchorage and insertion of the amphipathic helix. This can be seen also in Fig. 6c, d where the helical wheel of the helix is shown. The Ser residue is the single polar residue at the hydrophobic side of the amphipathic helix, and its change to Leu will have as a consequence to anchor the helix stronger to the membrane (Fig. 6c, d). Consistent with the known regulatory function of this segment in flaviviruses[41–46], we propose that the E S455L substitution modulates the membrane fusion process and/or E protein membrane incorporation. Thus, albeit future studies are required to demonstrate this hypothesis, the structural property of the E S455L substitution is in accordance with improved ability to initiate *de novo* infection (i.e., increased specific infectivity) of the E adaptive mutant.

## Discussion

Experimental evolution in a stable and controlled cell environment uncovered ZIKV adaptations leading to the avoidance of

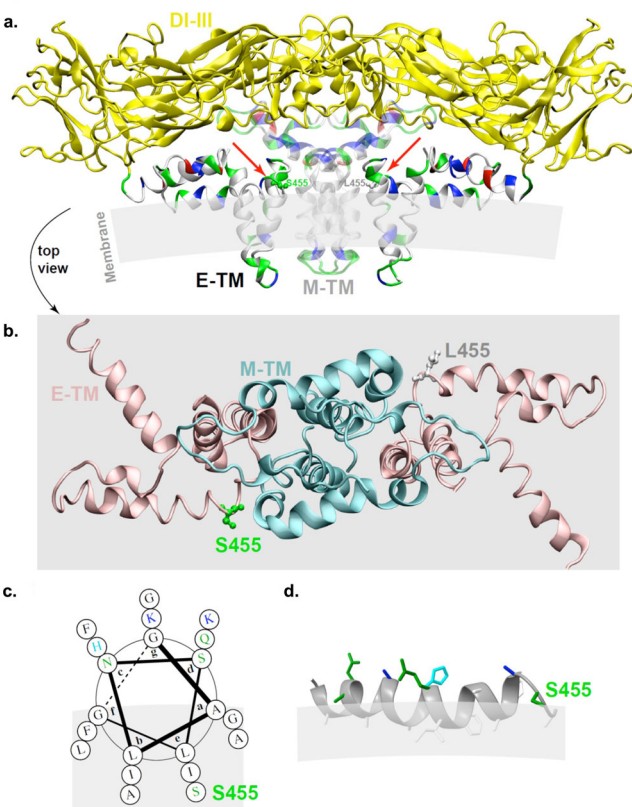

**Fig. 6 Positioning of the S455L mutation in E structure. a** Structure of ZIKA E (in solid) and M (transparent) proteins (5IRE.pdb)[42, 43],), with EI-III domains forming the envelope shell (yellow). The E and M protein transmembrane (TM) domains are shown with residue types indicated on the structure (white, hydrophobic; green, polar; blue, basic; red, acidic) in order to locate membrane insertion and to point out (red arrows) the positions of S455 (left monomer) and the mutant form L455 (right monomer) at the membrane interface. **b** Zoom of E-TM (cyan) and M-TM (pink), without EI-III, as viewed from the top of the structure shown in **a**. The positions of the wild-type S455 and mutant L455 residues at the interface between E and M TM domains are shown on one monomer each (the orientation of the L455 side chain is arbitrary), and has been created using the mutation tool in Swiss-PdbViewer. **c** Helical wheel (created with DrawCoil (http://www.grigoryanlab.org) of the helical segment with S455 indicated. The amphipathic helix is shown with the possible membrane bilayer in gray. **d** Side view of the helix with S455 located just above the membrane interface; mutation to L should drive deeper insertion of the helix into the membrane.

induced host antiviral response *via* an increased specific infectivity. Bioinformatic analysis of the viral genome evolution pinpointed two candidate mutations whose frequencies increased concomitantly with the acquisition of the adapted phenotype. We further showed that viral adaptation augmented infection independently of the ISG response and in specific cellular environments, thus suggesting a modulation of the host-virus interaction involved at the early step of ZIKV infection. Consistently, we uncovered a key determinant in the envelope (E) protein that both augments the probability for one physical virus to initiate infection in certain cell environments and mediates resistance to the TLR3-induced antiviral response.

**Proposed mechanism for the resistance to antiviral response.** Recent reports have shown the ability of *Flaviviridae* to acquire

adaptive mutations over the course of long passaging in cell culture[47,48]. Based notably on the sequencing of molecular clones, these studies concluded that adaptation to the host cell environment could occur through multiple, independent mutational pathways. In the present study, we found instead that in independent experiments the same mutations increased in frequency after a few passages in a controlled cellular environment. This difference between previous studies[47,48] and our results might be explained by a strong selective advantage provided by mutations present at a low frequency in the initial viral populations. Previous studies in the context of the infection by hepatitis C virus, known to lead to high diversity upon chronic infection in patients, showed that mutation frequencies underwent waves of increases and decreases during experimental evolution[47]; in some experiments, waves can be observed in the frequency of some variants (e.g., C2341T and C1786T; Fig. 3d–f).

Since the acquisition of both increased specific infectivity and resistance to TLR3-induced antiviral response were temporally associated during the experimental evolution, and knowing that the E S455L point mutation reproduces both phenotypes, we propose that the two phenotypes are causally linked. Like other flaviviruses, ZIKV has evolved inhibitory mechanisms against antiviral responses, including blockage of the response to type I and III IFNs by NS5-mediated degradation of STAT2[1–4]. Our results suggest that in our experiments viral adaptation occurred *via* a reduction of the time delay prior to the onset of productive infection rather than by directly modulating virus-mediated inhibition of the host antiviral detection. Firstly, we showed that improved viral replication of the adapted populations was independent of the amplitude of the ISG response, as demonstrated using HEK-293 where ISG upregulation is absent. Secondly, the comparison of the U6A-STAT2 cells versus the corresponding STAT2-deficient U6A cells indicated that the adapted viral populations did not differentially inhibit ISG responses induced by Jak/STAT signaling compared to the parental virus. Thirdly, kinetic analysis demonstrated that the adapted viral populations showed enhanced resistance to TLR3-mediated inhibition at later time-points post-infection, and with higher viral input. Fourthly, our mathematical model of ZIKV infection and replication showed that a reduced time-lag prior to the establishment of productive infection may suffice to cause increased virus spread, and hence associated resistance to the antiviral response. In accordance with the model, we demonstrated increased specific infectivity of the adapted viral populations and E S455L mutated recombinant virus, suggesting adaptation of host-ZIKV interaction at the early step of infection. According to our hypothesis, the resistance to TLR3-induced antiviral response is a coincidental result of increased specific infectivity that is the phenotype associated to a selective advantage during experimental evolution.

As this viral adaptation is cell type-restricted, the adapted viral populations likely become fitter by interacting with host factors preferentially present or conversely at a limiting level in certain cell types at early stages of infection. These host factors can include some machineries of the entry pathway such as endosome acidification and/or expression of cell type restricted receptor (e.g., DC-sign primarily expressed in dendritic and myeloid cells)[16,49–51]. Overall, our results indicate that the observed viral adaptation most likely results from an improved ability to rapidly and efficiently establish viral replication. Hence, faster accumulation of viral products in newly infected cells can overwhelm and/or bypass the host antiviral factors.

**Identification of a key determinant in ZIKV E responsible for increased specific infectivity.** Bioinformatic analysis identified

the mutations S455L in E and Y87H in NS4B as reproducibly associated to the adapted phenotype. These mutations were at low frequency at the beginning of the experiment; then, presumably strong fitness benefits allowed them to rise in frequency through selection. The functional analysis by insertion of the candidate mutations into a ZIKV molecular clone demonstrated that the S455L substitution in E controls both improved viral spread and resistance to TLR3-induced antiviral signaling. According to our mathematical model, the faster replication onset of the mutated virus could outpace the host response. Therefore, this viral adaptation may also confer resistance to other innate signaling pathways. Demonstrating this hypothesis would require further investigation.

The cellular study model has reduced complexity compared to whole organisms (e.g., unique cell type, restricted diversity of antiviral sensing and absence of adaptive response and physico-chemical constraints such as blood flow). This enabled the discovery of viral adaptation improving fitness in a stable and well-defined cell environment[52,53]. At the organism scale, interferences between several selective pressures could impede the acquisition of some optimized interaction(s). This may explain why the adaptive mutation at position 455 of the E envelope protein is found only in 2 out of 519 ZIKV full genome sequences isolated from patients (NCBI Virus Variation Resource on 21 December 2020) and is not maintained in a mutational scanning using a different cell type and strain[54]. Alternatively, adaptive mutations may enable the virus to infect different cell types with differential efficiency. In this scenario, the most frequent ZIKV variants would be well adapted to infect the major target cell types, while low-frequency variants would be better at infecting specific cell types. In accordance with this, comparison between various cell types showed that the adapted phenotype is cell type-restricted, indicating that the targeted host factor(s) is differentially involved in viral replication depending on the cell type. This also suggests cell type-specific trade-offs for the virus: adaptation to a particular cell environment limits infectivity in other cellular environments. Consistently, we observed that the adapted virus had a reduced ability to infect the monocyte-derived macrophages, possibly as a result of different cell entry mechanisms. Such trade-offs are in line with the results of Duggal et al.[55] who found that mutations appeared during passages in Vero cells reduce the pathogenicity of the virus in mouse; it is unclear whether differences in cell types or in host species underlie this observation.

Based on analysis at the structure level, we propose that the E S455L substitution modulates the membrane fusion process and/or E protein membrane incorporation. Future studies are required to test this hypothesis, yet the expected structural effect of the E S455L substitution is consistent with the demonstration of an ability to better initiate de novo infection (i.e., increased specific infectivity) of the E adaptive mutant. Therefore, we propose a working model of viral adaptation via an optimized interaction with the host machinery involved at an early stage of infection, via E membrane interactions, and likely by modulating its function in membrane fusion. In turn, higher infectivity would result in the accelerated accumulation of viral products and/or altered entry pathway that overcome or bypass host antiviral responses.

In conclusion, we studied viral adaptation in a stable and controlled cellular environment leading us to link phenotype to mutation. The methods of deep sequencing and bioinformatics set up here allowed the identification of mutations at low frequency arising across the entire viral genome, as expected for RNA viruses with a high error rate during replication. The profiles of frequency increase of the variants suggested that they were bona fide variants, not experimental artifacts of our sequencing protocol. Focusing on the mutations that reached high frequency concomitantly with the adapted phenotype, we functionally validated one of them by recapitulating the observed phenotypes, modeled how its effect on viral replication could result in an improved fitness of the mutant, and examined its likely impact on protein structure. Overall our approach allowed us to combine viral evolution, mathematical and structural modeling as well as functional assays in order to better understand viral adaptation.

## Methods

**Biological materials**. Huh7.5.1 cells (kindly provided by Dr F.V. Chisari; Scripps Research Institute), Vero E6 cells (kindly provided by Dr M Bouloy; Institut Pasteur) and HEK-293 cells (ATCC CRL-1573) were maintained in Dulbecco's modified Eagle medium (DMEM) supplemented with 10% Fetal Bovine Serum (FBS), penicillin (100 units per mL; (U)/mL), streptomycin (100 µg/mL), non-essential amino acids (100 nM) and Hepes (10 mM) at 37 °C/5% $CO_2$, as previously described[56]. The fibrosarcoma U6A cells (kindly provided by Dr M. Köster; Helmholtz-Zentrum für Infektionsforschung) expressing or not human STAT2[57] were cultured in DMEM supplemented with 10% FBS, penicillin (100 U/mL), streptomycin (100 µg/mL), non-essential amino acids (100 nM), hepes (10 mM), and sodium pyruvate (1 mM) at 37 °C in 5% $CO_2$. All cell culture reagents were purchased from Life Technologies. Monocytes were isolated from blood from healthy adult human volunteers, obtained according to procedures approved by the 'Etablissement Français du sang' (EFS) Committee. PBMCs were isolated using Ficoll-Hypaque density centrifugation and monocytes were positively selected using anti-CD14 microbeads (MACS Miltenyi Biotec) according to the manufacturer's instructions. Then, 2x10e5 CD14 + cells were differentiated into monocyte-derived macrophages in RPMI 1640 culture medium with 10% FBS, penicillin (100 U/mL), streptomycin (100 µg/mL), non-essential amino acids (100 nM), hepes (10 mM), sodium pyruvate (1 mM), 0.05 mM of β-mercaptoethanol, Glutamine (2 mM), and supplemented with 100 ng/mL human M-CSF cytokine (Peprotech). Three days post-isolation, half of the media was refreshed by similar media supplemented with 100 ng/mL human M-CSF cytokine. Six days later, the differentiation was assessed by FACS analysis. The infections of macrophages derived from monocytes were performed at day 7 post-isolation. The U6A and HEK-293 cells were infected by the indicated viral populations with MOI of 0.1. Primary macrophages derived from monocytes were infected with MOI of 0.2.

To establish a controlled set up for the evolutionary experiment and in line with our strategy to work in a cell environment non-responsive to viral products, we used Huh7.5.1. In addition, to enable the subsequent study of the influence of activated antiviral response on the evolved ZIKV populations, such as TLR3-induced signaling by poly(I:C) treatment, we transduced Huh7.5.1 cells with a WT TLR3 and, as reference control, a mutant TLR3 invalidated for signaling via a deletion of the Toll/interleukin-1 receptor (TIR) domain of the cytosolic tail (ΔTIR-TLR3)[29,33]. The retroviral-based vectors expressing WT TLR3 and ΔTIR-TLR3 were produced in HEK-293T cells and used to transduce Huh7.5.1 cells, as we previously described[58]. Transduced cells were selected by adding blasticidin at 10 µg/ml (Invivogen).

**Reagents**. The antibodies used for immunostaining were mouse anti-E glyco-protein (4G2) kindly provided by P. Despres (PIMIT, Université de La Réunion-INSERM France) and anti-Mouse IgG (H + L) Alexa Fluor 555 Secondary Antibody (Life Technologies). The antibodies used for western blot detection were anti-ISG15 (Santa Cruz), anti-flag (clone M2; Sigma-Aldrich) and mouse anti-actin (clone AC74; Sigma-Aldrich). The antibodies used for FACS analysis were APC-conjugated anti-CD163 (clone REA812, MACS Miltenyi Biotec), PE-conjugated mouse anti-HLA-DR (clone LN3, Invitrogen); PE-Cy7-conjugated mouse anti-CD11b (clone ICRF44, BD Biosciences), Pacific Blue-conjugated mouse anti-CD14 (clone 63D3, Biolegend). Other reagents include TLR3 agonist poly(I:C) (Invivogen); Hoescht (Life Technologies) and cDNA synthesis and qPCR kits (Life Technologies). The reagents for the preparation of the libraries include: Vivaspin unit 100 MWA (Sartorius); Pico Total RNA kit (Agilent), T4 RNA ligase (New England Biolabs); Superscript III (Life Technology); Ultra DNA Library Prep kit for Illumina (New England Biolabs): AMPure XP beads (Agencourt); and High Sensitivity DNA kit (Agilent).

**Analysis of intracellular and extracellular RNA levels**. RNAs were isolated from cells or supernatants harvested in guanidinium thiocyanate citrate buffer (GTC) by phenol/chloroform extraction procedure as previously described[56]. The efficiency of RNA extraction and reverse transcription-real-time quantitative PCR (RT-qPCR) was controlled by the addition of carrier RNAs encoding Xef1α (Xenopus transcription factor 1α) in vitro transcripts in supernatants diluted in GTC buffer. ZIKV RNA, Xef1α and intracellular MxA, ISG15, ISG56, and glyceraldehyde-3-phosphate dehydrogenase (GAPDH) mRNA levels were determined by RT-qPCR using cDNA synthesis and qPCR kits (Life Technologies) and analyzed using StepOnePlus Real-Time PCR system (Life Technologies), using previously

described primers[31]. Primers were designed to the region of prM protein (nucleotide positions: 689-to-782) of the ZIKV genome and are described in the Supplementary Table 1. Extracellular and intracellular ZIKV RNA levels were normalized for Xef1α and GAPDH RNA levels, respectively.

**Analysis of extracellular infectivity.** Infectivity titers in supernatants were determined by end-point dilution using Huh7.5.1 cells. Foci forming unit (ffu) were detected 48 h after infection by anti-E envelope protein specific immuno-fluorescence using 4G2 clone, as previously described[31]. Briefly, Huh7.5.1 cells were fixed with 4% PFA and permeabilized by incubation for 7 min in PBS containing 0.1% Triton. Cells were then blocked in PBS containing 3% BSA for 15 min and incubated for 1 h with mouse anti-E (clone 4G2) hybridoma supernatant, 1:100-dilution in PBS containing 1% BSA. After three washes with PBS, cells were incubated 1 h with secondary Alexa 555-conjugated anti-mouse antibody (1 μg/mL) and Hoechst dye (200 ng/ml) in PBS containing 1% BSA. Foci countings were performed using Zeiss Axiovert 135 or Olympus CKX53 microscopes.

**Determination of the specific infectivity.** The specific infectivity was determined as the ratio of the extracellular infectivity titer value (determined by ffu counting) relative to the extracellular ZIKV RNA value (as absolute number genome equivalent quantified by RT-qPCR) for each individual experiment, and expressed as the mean values ± standard deviation (SD).

**Foci size analysis.** Foci sizes were determined by infection using Huh7.5.1 cells and Vero cells with a carboxymethylcellulose (CMC) overlay in 12-well plate. Forty-eight hours later, cells were washed with PBS and fixed by 4% PFA. Foci were detected by anti-E protein (4G2 clone), as described above. Imaging by Olympus CKX53 microscope of randomly picked foci were processed using Image J software. Size index was determined as the ratio of E staining intensity relative to Hoechst staining with normalization of each intensity on the mean intensity of three randomly selected cell units. More than 80 foci were acquired for parental and passaged virus comparison, and more than 50 foci for the molecular clone analysis.

**Immunostaining and FACS analysis of monocyte-derived macrophages.** Surface immunostainings were performed at day 6 post-isolation of the CD14 + cells from PBMCs. After a 15 min-incubation step with human Fc blocking reagent (MACS Miltenyi Biotec), cell surface markers were detected by a 30 min-incubation at 4 °C with 4 μg/mL Pacific Blue-conjugated mouse anti-CD14 (clone 63D3, Biolegend), a 1:20 dilution of APC-conjugated mouse anti-CD163 (clone REA812, MACS Miltenyi Biotec), 0.14 μg/mL PE-conjugated mouse anti-HLA-DR (clone LN3, Invitrogen) and a 1:20 dilution of PE-Cy7-conjugated mouse anti-CD11b (clone ICRF44, BD Biosciences), diluted in staining buffer (PBS - 1% FBS). Flow cytometric analysis was performed using a BD FACS Canto II and the data were analyzed with Flow Jo software (Tree Star).

**Western blot analysis.** Cell lysates of the indicated cells were extracted using lysis buffer (150 mM NaCl 50 mM Tris HCl pH 8, 1% NP40, 0.5% Deoxycholate, 0.1% Sodium dodecyl sulfate) and analyzed by Western blotting using anti-ISG15 (sc-50366, H150; Santa cruz; 1 μg/mL), anti-Flag (clone M2 Sigma-Aldrich; 1 μg/mL), and actin (clone AC74; Sigma-Aldrich; 1 μg/mL), followed by secondary HRP-coupled antibodies and chemiluminescence.

**Serial passaging of viral populations, deep sequencing and bioinformatic analysis for the selection of variants.** A clinical isolate of ZIKV from Brazil collected from a patient during the epidemics (PE243_KX197192) was amplified using Vero E6 cells[39]. Supernatants were harvested and filtrated (0.45 μm) before storage at -80 °C. The serial passaging of viral populations was performed by inoculation of Huh7.5.1 cells expressing ΔTIR TLR3, or WT TLR3 pre-treated with poly(I:C). The target cells were seeded at 2×10⁶ cells in two T175 cell culture flasks (Corning) the day prior infection (i.e., at MOI 0.1 for the first infection, and at MOI 0.01 for subsequent viral passaging because of the limited viral yield from early passages, and MOI 0.1 for passage 6-to-18 of round #1). At 3 days post-infection, the supernatants were harvested and their infectivity was determined by foci counting. Next, harvested supernatants were used to infect naïve cells, which were similarly modified. The serial passaging of viral populations was repeated up to 18 times (for a total up to 54 days). To avoid contamination of the libraries by cellular RNA released from dying cells, an additional amplification step was performed by passaging of the supernatant with the corresponding cells for 2 days, using MOI of 0.1. Supernatants (SN, 10 ml, approx. 10E7 secreted infectious particles per sample were used) were concentrated using Vivaspin units with 100 MWA cut-of (Sartorius) by centrifugation at 3000×g for 20 min at room temperature. Viral RNAs were extracted from the concentrated SN by phenol-chloroform extraction procedure as previously described[58]. Next, RNAs were fragmented by sonication using Covaris M220 (peak incident 50 Watts; duty cycle 20%; 200 cycles per burst, time 200 seconds at 4 °C) using Covaris M220. Fragmented RNAs were concentrated with isopropanol precipitation, followed by analysis with Bioanalyzer (Agilent Technology) using Pico Total RNA Chip. The median size of the RNA fragments was around 150 nucleotides. We then generated

tandem repeats of the fragment to reduce the error rate of next-generation sequencing[25]. The steps of circularization of RNA fragments and retrotranscription were adapted from a previously described protocol[25]. Briefly, circular RNAs were obtained by ligation using T4 RNA ligase (New England Biolabs), followed by phenol-chloroform extraction. The retrotranscription was performed using Superscript III (Life Technology). Then, libraries were prepared using the Ultra DNA Library Prep kit for Illumina (New England Biolabs). Additional steps of clean-up were performed using AMPure XP beads, ratio 1:0.8 (Agencourt) to remove free adapters (done before the step of PCR enrichment of adapter-ligated DNA) and to remove the free index adapters (after the PCR enrichment of adapter-ligated DNA). The quality of the libraries was assessed by Bioanalyzer (Agilent Technology) using High Sensitivity DNA Chip. The libraries were quantified using NEB Next Libraries Quantification kit for Illumina (New England Biolabs) and multiplexed at equimolarity. Multiplexed libraries were sequenced using HiSeq 2500XL (Illumina), using a 200-PE run at the EMBL Genecore Facility (Heidelberg, Germany). Given the median fragment size of 150 bases, at least two repeats are expected in most fragments, allowing to reduce the theoretical minimum error rate associated with current Illumina sequencing from $10-4$ to $10-8$ per base.

**Bioinformatic analysis of the sequencing libraries.** Reads were quality-checked and trimmed of sequencing adapters and then mapped using PEAR[59] and in-house software (https://github.com/Kobert/viroMapper). PEAR is an efficient and flexible tool that was used to merge the repeats in the forward and reverse reads, thus generating consensus sequences with improved quality scores and reduced sequencing errors. Viromapper automatically returns a table with statistics per position such as quality quantiles, coverage, raw numbers of A, C, G, T bases, and quality-weighted counts and proportions of A, C, G, T bases. These tables were then read and analyzed using Python scripts developed in house. To evaluate the efficiency of our protocol to generate tandem repeat, we measured repeat sizes within forward and reverse reads separately using a script developed in house, based on samples of 5000 reads in 22 read files. All libraries contained repeats. The worst library (HV5GLBCXY_ZIKV_17s006139-1-1_DREUX_lane1CirseqD3) had repeats in 59% of its forward reads and 67% of its reverse reads, with sizes ranging between 33 and 97 bases, with a mean at 50. In all other libraries, repeat sizes had similar distributions, but repeats were present in up to 94% of the reads. Datasets for the sequence data file available in SRA with accession numbers SRX9704326-SRX9704344; bioproject PRJNA686429.

**Bioinformatics analysis for the selection of variants of interest.** We were interested in variants absent or at low frequency in the starting clone and that reached high frequencies during the experiments. We reasoned that the variants of interest should vary in frequency in the course of several independent experiments. To detect variants of interest, for each site and in each experiment, we computed the standard deviation of the frequency of the major variant. The distribution of these standard deviations is represented in Supplementary Fig. 4b, c. Based on these standard deviations, we counted the number of experiments in which a given site had a standard deviation above some threshold. Two threshold values were used: 0.02, and 0.1 (Supplementary Fig. 4). Sites that passed the threshold in several experiments were examined further (Supplementary Fig. 4). A Jupyter notebook reproducing these analyses is available (https://github.com/Boussau/Notebooks/blob/master/Notebooks/Viromics/FiguresAndAnalyses_NoTLR3.ipynb).

**Introduction of selected mutation and analysis of adapted mutants in the ZIKV infectious clone.** Mutations were introduced in the genomic length ZIKV infectious clone cDNA, pCCI-SP6-ZIKV BeH819015 plasmid (kindly provided by Dr A. Merits)[40]. For adequate comparison with clinical isolate PE243[39], the R99G and Y122H substitution in NS1 were also introduced in pCCI-SP6-ZIKV BeH819015 plasmid.

In a first step, introduction of Y87H in NS4B were performed by overlap-PCR (OL-PCR) using mutagenic primers, as described previously[56]. The OL-PCR fragments were purified and transferred using In-Fusion HD Cloning Plus CE technology (638917, Ozyme) into the pCCI-SP6-ZIKV BeH819015 plasmid cleaved using BamHI restriction enzyme.

In a second step, the S455L substitution in E protein and R98G and Y121H substitution in NS1 were obtained by generating a synthetic gene i.e., fragment from nucleotides 1507-to-3375 from 5'UTR of BeH819015 sequence, containing the NS1 mutations combined or not with E S455L substitution. Next, the different fragments were amplified by PCR and transferred by using In-Fusion HD Cloning Plus CE technology into the pCCI-SP6-ZIKV BeH819015 plasmid which contained or not the Y87H substitution in NS4B that were cleaved/opened using Avr2 and Pml1 restriction enzyme.

All the sequences of the primers used for the mutagenesis and In-Fusion HD Cloning Plus CE technology are described in the Supplementary Table 1. The expected sequences for the PCR derived regions and the presence of mutations were validated by Sanger sequencing.

**Analysis of adapted mutants in the ZIKV infectious clone.** In vitro RNA transcripts were prepared from the parental pCCI-SP6-ZIKV BeH819015 plasmids and the above-described mutated ZIKV BeH819015, including the S455L and/or

Y87H substitution. Briefly, these plasmids were linearized with AgeI. After DNA extraction, in vitro RNA transcripts were generated using mMESSAGE mMA-CHINE SP6 Kit (Ambion) followed by Lithium precipitation, as previously described[56]. For the experiments presented in Fig. 4 and Supplementary Fig. 6, in vitro RNA transcripts were transfected into Huh7.5.1 cells using Lipofectamine 3000 transfection reagent (Life Technologies), following the manufacturer instruction. At 24, 48 and 72 h post-transfection, the supernatants were collected for the quantification of viral RNA and infectious titer. At 6 h post-transfection, the cells were washed with PBS and harvested to determine RNA levels.

### Statistics and reproducibility

*Statistical analysis of viral parameters*. Data are presented as the mean values ± standard deviation (SD). The Figure legend section reports the number of independent experiments. Statistical analysis was performed using R software environment for statistical computing and graphics, version 3.3.2. For levels of viral RNA and specific infectivity, the values were considered relative to reference/parental for each independent experiment and analyzed using an one-way ANOVA on ranks (Kruskal−Wallis test), as we previously performed[31]. When the test was significant, we used the Nemenyi *post hoc* test for pairwise comparisons of mean rank sums to determine which contrast(s) between individual experimental condition pairs was significant. The *p*-values from the Tukey Kramer (Nemenyi) pairwise tests are indicated in the table at the top of the graphs, when $p < 0.05$. Statistical analysis of foci size was performed by pairwise comparisons using Wilcoxon rank sum test. Statistical analysis of the kinetics of viral replication of the different viral populations was done using a mixed linear model, which takes into account the levels at the different time-point before those indicated in the table. Data considered significant demonstrated adjusted p-value by False Discovery Rate (FDR) <0.05, as previously[31].

*Mathematical modeling*. The multi-scale model of virus replication is based on a coupled system of partial and ordinary differential equations. Statistical evaluation of best-fit parameters and 95% confidence intervals were based on maximum likelihood estimation. For details see Supplementary note 1.

**Reporting summary**. Further information on research design is available in the Nature Research Reporting Summary linked to this article.

### Data availability

Datasets for the sequence data file are available in SRA with accession numbers SRX9704326-SRX9704344; bioproject PRJNA686429. A Jupyter notebook reproducing bioinformatic analyses is available at https://github.com/Boussau/Notebooks/blob/master/Notebooks/Viromics/FiguresAndAnalyses_NoTLR3.ipynb. Uncropped and unedited gel images for Supplementary Fig. 1 are provided as Supplementary Fig. 9.

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

## Acknowledgements
We thank P. Despres (PIMIT, Université de La Réunion-INSERM France) for the anti-E 4G2 antibody; F.V. Chisari (Scripps Research Institute, La Jolla, CA) for Huh-7.5.1 cells; Dr M Bouloy (Institut Pasteur, Paris, France) for Vero E6 cells and by Dr M. Köster (Helmholtz-Zentrum für Infektionsforschung) for U6A cells. We are grateful to Y. Jaillais for critical reading of the manuscript. We acknowledge the contribution of SFR Biosciences (UMS3444/CNRS, US8/Inserm, ENS de Lyon, UCBL) facilities. This work was supported by grants from the "Agence Nationale pour la Recherche" (ANR-JCJC-EXAMIN), the "Agence Nationale pour la Recherche contre le SIDA et les Hépatites Virales" (ANRS-AO 2017-01), the European Union's Horizon 2020 Research and innovation program under "ZIKALLIANCE" (Grant Agreement no. 734548) and FINOVI foundation (AO11) to M.D. The grants from "Fondation pour la recherche médicale" (contract Bioinformatic analysis for research in biology, DBI20141231313 and from the "Agence Nationale pour la Recherche" LabEx Ecofect (Grant ANR-11-LABX-0048) to M.D., B.B., and A.B.; A.K. is supported by the UK Medical Research Council (MC_UU_12014/8, MR/N017552/1).

## Author contributions
V.G., T.H., B.B., and M.D. designed research; V.G., E.H., K.K., S.R.T., E.D., C.G., P.V.M., M.P., S.M.-G., and L.S. performed research; V.G., E.H., K.K., S.R.T., E.D., C.G., P.V.M., A.K., A.B., M.P., S.M.-G., and L.S. contributed new reagents/analytic tools; V.G., E.H., K.K., S.R.T., E.D., C.G., P.V.M., A.K., M.P., A.B., S.M.-G., L.S., T.H., B.B., and M.D. analyzed data; and V.G., B.B., and M.D. wrote the paper.

## Competing interests
The authors declare no competing interests.
