## [Peer Review File · Communications Biology]

Reviewers' comments:

Reviewer #1 (Remarks to the Author):

Grass, et al provide a detail analysis of the mutations that accumulate in passage of Zika virus in cell culture. The authors report the experimental evolution and sequencing of ZIKV in human-derived Huh7.5.1 cells. The authors use a mix of virology, cell biology and modeling to characterize the phenotype of a dominant mutation that emerged during passage.

Let me first say that I appreciate this paper for its use of a combination of quantitative methods to characterize the mechanisms of viral evolution. I think with some revisions, the paper could be suitable for publication at Communications Biology.

General comments:

The authors perform selection in Huh7.5.1 cells and find the dominant mutation increases virus specific infectivity (as titer/extracellular RNA). They explore its ability to overcome TLR immune activation, finding some modest differences in TLR3 sensitivity.

1. It seemed odd that the evolutionary experiment wasn't done in both Huh7.5.1 and Huh7.5.1+TLR3 cells. Is the diversity different in these environments? Is the S455 mutation selected and/or retained after passage in TLR+ cells?

2. Is it meaningful that this confers TLR3 activation, or is this a correlate of other phenotypes that confer increased fitness in Huh7.5.1? Is it coincidence, pre-adaptation, epistasis?

3. The authors see induction of ISGs (e.g. MxA in Fig 1D) and a reduction in the adapted virus, could the phenotype confer resistance to other innate recognition?

I thought Figure S7 was particularly helpful in providing a bit of molecular insight into the effect of the E S455L mutation. I would suggest including some of that detail in the main manuscript beyond the final section of the discussion.

Specific comments:

II. 88-91: CirSeq.

The CirSeq method relies on the alignment of three tandem repeats and determination of majority rule consensus to generate the corrected sequence. How does the method used, PEAR, compare to the methodology and quality improvements of CirSeq? Perhaps the authors could comment briefly on this in the manuscript. Later, the authors chose a rather conservative cut-off for mutations (i.e. where the SD of frequencies across passage are >0.1) rather than the <10e-5 sensitivity of CirSeq. Is anything gained by looking at more rare SNVs? How much better is this than simple RNAseq and variant calling?

I. 123: "specific infectivity". The specific infectivity is computed as extracellular titer over the extracellular RNA. Is there a benefit to comparing infectivity to RNase-resistant GE's or purified particles (by sucrose cushion, for example)?

LI 178: Foci size could also reflect a higher number of infectious particles, not only replication speed?

Figure 1: Statistical analyses are not explained sufficiently.

LI. 909: Eqn. 6, Lipofectamine and Fig. 5B. I do not see a figure 5B in the manuscript, and no other mention of lipofectamine.

LI 111: What is the MOI of the virus at each passage? This is a critical parameter of these types of evolutionary experiments, but as best I can tell it is not mentioned in the text.

Figure 1A. I find the curved lines distracting and confusing. I would recommend removing them. S1B in the Supplementary data does not show any data points.

Figure 1D-H. I think a bit more detail on how the p-values are computed is needed here. Unclear what the hypothesis being tested is. Is described later on in the "Statistical Analysis of Viral Parameters" but should be made clear this is specifically looking at the intervals between timepoints and not the specified time points themselves (i.e. "24h-48h" vs "24h")

Figure 2. I found the the phrase "quantification of the frequency of the second most variants at each position" to be confusing. I gather that this is to exclude the dominant parental genotype. Perhaps saying that more directly would be best.

Figure 4. While this is not my area of expertise, I am concerned about the modeling results. Given the simplicity of the data (3 measurements), the parapets and the results, the model looks over-fit. Perhaps showing a bit more information about the error associated with the simulations would be helpful in evaluating the model's utility. The authors assertion that there are a "well constrained number of parameters" is not backed up by any quantification. Can alternative explanations of the adaptation be examined using this approach?

Virus is passed in Huh7.5.1 cells, and viruses adapts to TLR3 activation? When TLR3 activation is inhibited?

Reviewer #2 (Remarks to the Author):

Here, the authors have produced a paper investigating Zika virus infection dynamics during adaptation to culture in a human cell line. This was accomplished using experimental evolution by serially passaging Zika virus populations in Huh 7.5.1 cells and comparing phenotypic and genotypic parameters of input versus passaged viruses. Reverse genetics was then used to generate mutant viruses that differed only at codon E 455 and NS4B 87 in an otherwise isogenic background. This was done on the same Brazilian Zika virus backbone that was used for the serial passaging experiment. Finally, a mathematical model was generated that suggested that the enhanced replicative capacity of passaged virus populations may be the result of overall faster replication thus facilitating escape from the antiviral response. This is a comprehensive study but in places it is very difficult to follow. As a result, I believe there are several things that need to be addressed to facilitate understanding by the reader.

Line 106: I am not sure I understand what the authors mean by "a cell environment that is non-responsive to viral products". Also, what is the biological relevance of using a hepatocyte-like cell line?

Line 110: what is the diversity in the starting inoculum? Did you sequence your stock virus? (I see this is included later in the text but it would be worthwhile to mention the diversity in the stock inoculum.

Line 115 and throughout: the use of days to represent distinct passages is confusing. Why not just use passage 1-18? The number of days is irrelevant, because as I understand it after 3 days the virus was harvested, titrated, and a new infection was started at an MOI of 0.01 PFU/cell in new cell cultures.

Line 121-123: specific infectivity is the ratio of infectious virus to the total amount of vRNA, which is what is shown in Fig 1A but not how it is defined in the text.

Fig 1A: why do lineages 2 and 3 abruptly end before lineage 1?

Line 146-148: how do you explain the peaks and valleys in your specific infectivity curves? Further, it looks like the PFU:particle ratio is 1:1 at the 3 day timepoint, is that correct, and the same as with the stock?

Fig 1: why only show vRNA levels here and not infectious virus?

Line 239: it is not accurate to state that next generation sequencing cannot detect low frequency minority variants.

Line 258: Are there any minor variants present in your stock virus?

Figures: Much of the supplemental data are critical for interpretation of the results. I strongly suggest re-organizing the text and figures to logically incorporate supplemental data in the main text. In addition, I recommend revising figures to make them more intuitive and easier to read. In addition, I recommend splitting the figures into smaller multi panel figures that are inline with what is described in the text.

Referee expertise:

Referee #1: virology, evolution

Referee #2: virology, emerging infectious diseases

Point-by-point response to the Reviewers

We thank both Reviewers for their helpful comments, which we have addressed by including new results, rearranging the set of Figures and in-depth edition of the text.

The new results include:

- Analysis of the viral evolution in serial viral passages upon activated TLR3-signaling (**New Supplementary Fig. 5**)
- Additional molecular insights on the S455L mutation in the structure of E viral protein (**New Panels c-d in Fig. 6**).
- Additional analyses of the estimates of the parameters for the mathematical model, presented in **Revised Supplementary Fig. 7**.

Reviewers' comments:

Reviewer #1 (Remarks to the Author):

Grass, et al provide a detail analysis of the mutations that accumulate in passage of Zika virus in cell culture. The authors report the experimental evolution and sequencing of ZIKV in human-derived Huh7.5.1 cells. The authors use a mix of virology, cell biology and modeling to characterize the phenotype of a dominant mutation that emerged during passage.

Let me first say that I appreciate this paper for its use of a combination of quantitative methods to characterize the mechanisms of viral evolution. I think with some revisions, the paper could be suitable for publication at Communications Biology.

(Reply) We thank this Reviewer for pointing out the multidisciplinary of our study and for her/his helpful suggestions, which motivated the addition of new results.

General comments:

The authors perform selection in Huh7.5.1 cells and find the dominant mutation increases virus specific infectivity (as titer/extracellular RNA). They explore its ability to overcome TLR immune activation, finding some modest differences in TLR3 sensitivity.

1. It seemed odd that the evolutionary experiment wasn't done in both Huh7.5.1 and Huh7.5.1+TLR3 cells. Is the diversity different in these environments? Is the S455 mutation selected and/or retained after passage in TLR+ cells?

(Reply) The Reviewer raises a valid point, and we have now included new results of experimental evolution of ZIKV performed in TLR3-activated Huh7.5.1 cells (**New Supplementary Fig. 5a-e**).

As expected, upon induced antiviral response, we found that viral production was low in the first passages, but increased after passage 4-to-6 (**New Supplementary Fig. 5a**), and reached levels comparable to the corresponding Huh7.5.1 cells (**New Supplementary Fig. 5a versus Supplementary Fig. 1a**). This observation suggests that the viral populations harvested at the later passaging resisted to the activation of TLR3-mediated antiviral pathway.

We have also sequenced the harvested viruses, using the same protocol/methodology as for the viral evolution performed in the corresponding Huh7.5.1 cells. The most significant mutations obtained in TLR3-activated cells are shown in the **New Supplementary Fig. 5a-e**. We found an increased frequency of two mutations of interest, *i.e.*, C2341T and T7173C (**new Supplementary Fig. 5c**) The other mutations were also similar to the ones identified in corresponding Huh7.5.1 cells, although the increase of their frequency was lower for the experiments performed in TLR3-activated Huh7.5.1 cells compared to the corresponding Huh7.5.1 cells. The description of these new results is now included in Results section (page 13, lines 312-319): *We performed similar runs of viral evolution experiment upon induction*

of the TLR3 antiviral response, and using Huh7.5.1 cells expressing WT TLR3 stimulated by the agonist poly(I:C) (**Supplementary Fig. 5a-b**). We sequenced the resulting viral populations in the 3 independent runs. Interestingly, we observed an increase of the frequency for the C2341T and T7173C mutations upon ZIKV passages in this experimental setting (**Supplementary Fig. 5c**). Together, our bioinformatic analyses identified two point-mutations in E and NS4B proteins that might be involved in the phenotype of viral adaptation observed during experimental evolution, both in the absence and in the presence of the TLR3 antiviral response.

2. Is it meaningful that this confers TLR3 activation, or is this a correlate of other phenotypes that confer increased fitness in Huh7.5.1? Is it coincidence, pre-adaptation, epistasis? (Reply) The mutated viruses have increased resistance to the TLR3 antiviral signaling, as shown in **Fig. 1b-c and 4d**. Our hypothesis is that the faster replication onset of the mutated viruses allows them to outpace the cell response, as presented in paragraph entitled “*Increased specific infectivity of passaged virus enables resistance to TLR3-induced antiviral response.*”, page 8, lines 169-178 of the Results section, revised text as follows: “*The resistance to TLR3-induced antiviral response can result from a faster onset of infection by the passaged viral populations. In such a scenario, ongoing replication before the establishment of a robust antiviral response in host cells would out-compete the latter. In support of this hypothesis, the replication rate of passaged viral populations was significantly faster as compared to the parental virus in the same infection set-up (MOI 0.05) and in the absence of TLR3-induced response (Fig. 1b, solid lines and comparing the slope from the time-point 24 hours to 48 hours post-infection). Similar observations were made for viral populations harvested at distinct late time points of the experimental evolution, including at passages 9, 11 and 12 (Fig. 1f). These observations suggest that the passaged viral populations could outcompete host antiviral response by an earlier replication onset post-inoculation.*”.

In accordance with the suggestion by this Reviewer, this hypothesis implies that the resistance to TLR3 antiviral signaling is a coincidental result of a faster replication onset. We have now added a sentence in the Discussion section (page 18, lines 437-439) to clarify this point: “*According to our hypothesis, the resistance to TLR3-induced antiviral response is a coincidental result of increased specific infectivity that is the phenotype associated to a selective advantage during experimental evolution.*”

3. The authors see induction of ISGs (e.g., MxA in Fig 1D) and a reduction in the adapted virus, could the phenotype confer resistance to other innate recognition? (Reply) We thank this Reviewer for pointing out a potential of broader impact of our study. As the resistance is associated to faster replication onset, which can outpace the host response, it is conceivable that this viral adaptation also confers resistance to other innate signaling. This point is of interest for a future project and is now mentioned as an open-question in the Discussion section (page 19, lines 457-460): ‘*According to our mathematical model, the faster replication onset of the mutated virus could outpace the host response. Therefore, this viral adaptation may also confer resistance to other innate signaling pathways. Demonstrating this hypothesis would require further investigation*’.

I thought Figure S7 was particularly helpful in providing a bit of molecular insight into the effect of the E S455L mutation. I would suggest including some of that detail in the main manuscript beyond the final section of the discussion.

(Reply) As suggested by this Reviewer, we have now included more molecular information of the E S455L mutation in the Result section (page 16, lines 374-389): ‘*The position 455 is located at the C-terminus of the stem region, next to the transmembrane domain of the E protein⁴¹⁻⁴³. In the structure of E (Sire.pdb reference)⁴³, the position 455 is located near the membrane surface, in an environment of nearly exclusively polar and hydrophobic residues (Fig. 6a-b). It is located at the end of an amphipathic helix which spans residues 436-454. Its*

side chain hydroxyl does not engage in interactions with the transmembrane helix of the M protein facing it, but is likely involved in lipid headgroup interactions. A mutation from the polar Ser to the very hydrophobic Leu in this position thus has the potential to markedly change the membrane anchorage and insertion of the amphipathic helix. This can be seen also in the **Fig. 6c-d** where the helical wheel of the helix is shown. The Ser residue is the single polar residue at the hydrophobic side of the amphipathic helix, and its change to Leu will have as a consequence to anchor the helix stronger to the membrane (**Fig. 6c-d**). Consistent with the known regulatory function of this segment in flaviviruses⁴¹⁻⁴⁶, we propose that the E S455L substitution modulates the membrane fusion process and/or E protein membrane incorporation. Thus, albeit future studies are required to demonstrate this hypothesis, the structural property of the E S455L substitution is in accordance with improved ability to initiate *de novo* infection (i.e., increased specific infectivity) of the E adaptive mutant.'

Accordingly, the results of the structure analysis are now presented in the main set of Figures as new **Panels c-d in the Fig. 6**.

Specific comments:

II. 88-91: CirSeq.

The CirSeq method relies on the alignment of three tandem repeats and determination of majority rule consensus to generate the corrected sequence. How does the method used, PEAR, compare to the methodology and quality improvements of CirSeq? Perhaps the authors could comment briefly on this in the manuscript. Later, the authors chose a rather conservative cut-off for mutations (i.e. where the SD of frequencies across passage are >0.1) rather than the <10e-5 sensitivity of CirSeq. Is anything gained by looking at more rare SNVs? How much better is this than simple RNAseq and variant calling?

(Reply) PEAR is a bioinformatic tool designed to merge forward and reverse reads, even if these reads contain repeats, *i.e.*, contain a portion of the sequence that has been sequenced several times. These features make PEAR very well suited to analyze tandem repeats arising from the Cirseq protocol and when the sequencing is done in paired-end reads, while the sequencing in the previous publications with CirSeq (*e.g.*, Acevedo, *et al.*, Mutational and fitness landscapes of an RNA virus revealed through population sequencing. Nature 505, 686-690) was done single-end, and could thus rely on a different bioinformatic pipeline. The quality improvements expected from this pipeline are the same as those expected from the original pipeline, and depend on the number of repeats. If there is a single copy, the minimum error rate is 10E-4; two copies: 10E-8, three copies: 10E-12, etc. To clarify the benefits of using PEAR, we have now edited the text, by adding the following sentence in the Material and Methods section: '*Bioinformatic analysis of the sequencing libraries*': '*PEAR is an efficient and flexible tool that was used to merge the repeats in the forward and reverse reads, thus generating consensus sequences with improved quality scores and reduced sequencing errors.*' (page 28, lines 660-662). Regarding the threshold that we used to identify mutations of interest, we indeed chose a conservative threshold because we wanted to examine mutations by hand, and perform functional assays. We were thus aiming for a low number of mutations to consider.

I. 123: "specific infectivity". The specific infectivity is computed as extracellular titer over the extracellular RNA. Is there a benefit to comparing infectivity to RNase-resistant GE's or purified particles (by sucrose cushion, for example)?

(Reply) The determination of specific infectivity is widely performed without treatment because additional steps could bring a bias to interpret the results. For example, the relative viral yield from procedure like centrifugation with sucrose cushion is subjected to variation owing to the difference of stability of the viral particles. Especially, this can be an issue as the herein identified key mutations are located on the surface protein.

L1 178: Foci size could also reflect a higher number of infectious particles, not only replication speed?

(Reply) The number Foci reflects the number of infectious particles that initiate infection within the targeted cell line. Distinctively, the size of the foci is a measurement of the ability of a single viral particle to propagate to the neighboring cells due to the use of CMC overlay *i.e.*, its size depends on how fast the virus can spread in a given amount of time.

To further rule out any impact of the number of foci, low MOIs were adjusted to obtain the same number of foci for both parental or passaged viruses. In these experimental conditions, the sizes were significantly different, as quantified in revised **Fig. 2b** and **4c**.

Figure 1: Statistical analyses are not explained sufficiently.

(Reply) The statistical analyses are now indicated in the Figure Legend section (in addition to the material and methods) and as follows:

- **Panel b:** *'The statistical comparison of intracellular ZIKV GE levels for the same viral population at the same time point post-infection between activated-TLR3 and control cells is performed using a one-way ANOVA on ranks (Kruskal–Wallis test), as in ref: Assil et al. 2019 (DOI: 10.1016/j.chom.2019.03.005). When the test was considered significant (p-values ≤ 0.05), we used the Tukey Kramer (Nemenyi) pairwise test as post hoc test for pairwise comparisons of mean rank sums to determine which contrasts between individual experimental condition pairs were significant. The p-values from the Tukey Kramer (Nemenyi) pairwise test are indicated in the table at the top of the graphs, when $p < 0.05$.'* (page 37, lines 836-843)

- **Panel c:** *'p-values from pairwise comparisons using Wilcoxon rank sum test are shown'* (page 37, lines 861-862)

- **Panel d-h:** *'The p-values of the statistical analysis of the kinetics are determined using a mixed linear model, which takes into account the levels at the time-points before those indicated in the table on the right side of the graphs. The indicated p-values are for the comparison of passaged viral population versus parental virus, when $p < 0.05$.'* (page 37, lines 868-871)

We also extend the description of the statistical analysis in the Materials and Methods section, as follows: *When the test was significant, we used the Nemenyi post hoc test for pairwise comparisons of mean rank sums to determine which contrast(s) between individual experimental condition pairs was significant. . The p-values from the Tukey Kramer (Nemenyi) pairwise tests are indicated in the table at the top of the graphs, when $p < 0.05$.* (page 31, lines 728-731)

L1. 909: Eqn. 6, Lipofectamine and Fig. 5B. I do not see a figure 5B in the manuscript, and no other mention of lipofectamine.

(Reply) We apologize for this mistake. As mentioned in the Materials and Methods section, the lipofectamine reagent serves to transfect the ZIKV infectious clone produced by *in vitro* transcription from pCCI-SP6-ZIKV BeH819015 plasmids (page 30, lines 715-718): *'For the experiments presented in Fig. 3a and Supplementary Fig. 6, in vitro RNA transcripts were transfected into Huh7.5.1 cells using Lipofectamine 3000 transfection reagent (Life Technologies), following the manufacturer instruction'*. They include the parental and mutants with the S455L and/or Y87H substitutions which are presented in **Fig. 3a** and **Supplementary Fig. 6**.

L1 111: What is the MOI of the virus at each passage? This is a critical parameter of these types of evolutionary experiments, but as best I can tell it is not mentioned in the text.

(Reply) All technical details regarding the passaging of viruses were mentioned in the Materials and Methods, more precisely in the section: *'Serial passaging of viral populations, deep sequencing and bioinformatic analysis for the selection of variants'*. To further clarify this point, we have now added a note in the text of the Result section as follows (page 5, lines

114-118): ‘*Experimental evolution were performed by serial passaging of ZIKV: at each passage (up to 18 passages), viral populations harvested at 3 days post-infection were used to infect naïve cells (Fig. 1a, schema on the left side), using MOI 0.1 for the first infection by parental virus and MOI 0.01 for subsequent viral passaging because of the limited viral yield from early passages (see ‘Material and Method’ section)’.*

(Reply) We have changed the display of the line for all concerned figures, including Figure 1a-b; **Supplementary Fig. 1a-b**, the **New Supplementary Fig. 5a-b**. We have also added marks indicating the determined value in **Fig. 1a-b, 3d-f, 4a and 4d**, and **Supplementary Fig. 1a-b, the New Supplementary Fig. 5c-e**

Figure 1D-H. I think a bit more detail on how the p-values are computed is needed here. Unclear what the hypothesis being tested is. Is described later on in the “Statistical Analysis of Viral Parameters” but should be made clear this is specifically looking at the intervals between timepoints and not the specified time points themselves (i.e. “24h-48h” vs “24h”)
(Reply) We thank the Reviewer for her/his comment and to point out the need for further explanation on the statistical analysis performed, which enables to consider the evolution over time by using a mixed linear model. As above-mentioned in a previous reply, this is now clarified in the Figure Legend section, as follows: **Panel d-h**: ‘*The p-values of the statistical analysis of the kinetics are determined using a mixed linear model, which takes into account the levels at the time-points before those indicated in the table on the right side of the graphs. The indicated p-values are for the comparison of passaged viral population versus parental virus, when $p < 0.05$.*’ (page 37, lines 868-871)

Figure 2. I found the phrase “quantification of the frequency of the second most variants at each position” to be confusing. I gather that this is to exclude the dominant parental genotype. Perhaps saying that more directly would be best.

(Reply) We clarified the Figure Legend section of New Fig. 3b-c, as follows: “*b-c Time-course quantification of the frequency of the non-parental variants at each position along ZIKV genome in the viral populations harvested in at least one representative independent run of evolution experiment (b) and in the viral populations harvested at passage 6 in 3 independent runs of evolution experiments (c). In both cases, the frequency of the second most frequent variant is shown.*” (page 38, lines 878-882).

Figure 4. While this is not my area of expertise, I am concerned about the modeling results. Given the simplicity of the data (3 measurements), the parapets and the results, the model looks over-fit. Perhaps showing a bit more information about the error associated with the simulations would be helpful in evaluating the model’s utility. The authors assertion that there are a “well constrained number of parameters” is not backed up by any quantification. Can alternative explanations of the adaptation be examined using this approach?

(Reply) We have now provided more information regarding the fit of our mathematical model: (1) Is the model overfitting the experimental data (i.e., is the model too complex, given the information in the data)? (2) How did we arrive at our conclusion that the model parameters are well-constrained? He/she also asks whether alternative explanations for viral adaptation can be studied.

Indeed, points (1) and (2) are mathematically closely related. If the model is of appropriate complexity for the available data, the model parameters should be constrained by the data, or, in technical terms, the model parameters should be identifiable. We applied a standard method for determining parameter identifiability, the profile likelihood method (Raue et al., Ref. 65). The result is depicted in **Revised Supplementary Fig. 7** and shows that all eight parameters have meaningful upper 95% confidence bounds, as we discuss in more detail as follows, while five parameters also have lower bounds. Hence the model parameters are well-constrained by the experimental data.

The key biological insight of the model is that the delay to productive replication, τ_A , is significantly shorter for the adapted or mutated strains, 1.6 (95% confidence interval: 0, 5.7) hours as compared to 8 hours for the parental and reference strains (fixed according to the experimental measurement by Zmurko et al. 2016, Ref. 59). Hence, the informative feature is here the upper 95% confidence bound. The non-existence of a lower bound does not affect the result that the delay for adapted/mutated strains is significantly smaller than the parental/reference strain delay of 8 hours.

The estimated intracellular virus replication rate (k_{rep}), which is bounded, matches single-cell measurements of poliovirus replication rate (Guo et al. 2017) and can thus be regarded as reasonable.

The upper bound of the virus production from intracellular viral RNA (k_{vp}) means that about 2 in 10^4 intracellular RNA genomes are packaged into an infective virus particles per hour. This number is determined by experimental measurements of the ratio of the extracellular infectivity ($\sim 10^4$ ffu/ml) to the intracellular ZIKV concentration (10^8 GE/ μ g total RNA; see **Fig. 3**).

With the estimated value of the infection rate (k_{inf}), the basic reproductive ratio of the ZIKV is ~ 3 , which is compatible with the report by Best and Perelson, 2018 (PMID: 30129207).

The estimated number of ZIKV RNA per cell upon infection (R_0) is in the order of thousand RNA molecules per cell. The 95% CI shows that this parameter is constrained by the data albeit not particularly well. Similar considerations hold for the maximum number of viral RNA per cell (R_{cc}). Importantly, the rather large 95% CIs for these parameters do not compromise the prediction of the lower τ_A for adapted/mutated strains (see above).

The estimated value of the death rate of the infected cell (k_{icd}) suggests a half-life between 4 and 20 hours. This is reasonable as an order-of-magnitude estimate. For example, Yang et al. 2020 (PMID: 33424801) have shown that ZIKV infection disrupts mitochondrial membrane potential within 24 hours post-infection.

The L2V factor converts the Lipofectamine-transfection to infecting the cell culture with [90 1340 2.4x10⁴] infective virus particles. This parameter is experimentally not known.

We have now further clarified the quality of parameter estimation in the text (page 15), focusing on our key conclusion regarding τ_A .

Given that the parameters are well-constrained, we expected that the model fit also has narrow confidence bounds, which is a further test of whether the model is appropriate for the experimental data or overfit. In particular, an overfit model would not be appropriately constrained by the data (including the measurement errors) and have large confidence bounds, although the best fit line recapitulates the data rather precisely. By contrast, a good model recapitulates the data within measurement error and has quite narrow confidence bands around the data. We now additionally computed the 95% confidence bounds of the model fit and show it in the **Revised Supplementary Fig. 7 and Fig. 5b-c**. Indeed, the confidence bounds are narrow, supporting that the model is of appropriate complexity. We amended the respective Figure Legend section and the **Appendix 1** describing the modeling, and also added a brief section to the Material and Methods section.

Finally, this Reviewer asked whether alternative explanations for the observed mutant phenotypes can be examined using mathematical modeling. Given concrete competing hypotheses, this may well be possible by statistical model selection. The purpose of the present model was to show that the experimental data are quantitatively consistent with a specific mechanism of viral adaptation – accelerated replication – that was implicated by the experiments. We hope that the additional material and clarification convinced this Reviewer that our modeling involves sound statistical evaluation of the estimated model parameters and model fits.

Virus is passed in Huh7.5.1 cells, and viruses adapts to TLR3 activation? When TLR3 activation is inhibited?

(Reply) We have now also added the results of sequencing data obtained when passaging the virus in activated-TLR3 Huh7.5.1 cells. Briefly and as above-mentioned in the response to

general comment point #1, we observed that the same main mutations arose in both cellular environments, and approximately at the same passages (*i.e.*, passage 4-to-6). Of note, one of these mutations, mutation E S455L, allowed resistance to TLR3 signaling responses. The results are presented in the **New Supplementary Figure 5**.

Reviewer #2 (Remarks to the Author):

Here, the authors have produced a paper investigating Zika virus infection dynamics during adaptation to culture in a human cell line. This was accomplished using experimental evolution by serially passaging Zika virus populations in Huh 7.5.1 cells and comparing phenotypic and genotypic parameters of input versus passaged viruses. Reverse genetics was then used to generate mutant viruses that differed only at codon E 455 and NS4B 87 in an otherwise isogenic background. This was done on the same Brazilian Zika virus backbone that was used for the serial passaging experiment. Finally, a mathematical model was generated that suggested that the enhanced replicative capacity of passaged virus populations may be the result of overall faster replication thus facilitating escape from the antiviral response. This is a comprehensive study but in places it is very difficult to follow. As a result, I believe there are several things that need to be addressed to facilitate understanding by the reader.

(Reply) We thank this Reviewer for pointing out the comprehensiveness of our study and her/his helpful suggestions to improve the understanding of the results.

Line 106: I am not sure I understand what the authors mean by “a cell environment that is non-responsive to viral products”. Also, what is the biological relevance of using a hepatocyte-like cell line?

(Reply) We have now edited the text of the Results section and further developed the justification for the selection of a hepatocyte-like cell line. We selected the Huh7.5.1 cell line in order to: 1) control innate immunity induction as these cells have been reported to have genomic mutations limiting an ISG response (*e.g.*, deletion of RIG-I) and a lack of other immune sensors (*e.g.*, TLR3), 2) to obtain a high yield of viral production in accordance with the methodology of deep sequencing of the infectious supernatants, and 3) to avoid excessive cytopathogenic effect caused by viral infections. This is now included in the paragraph ‘*Acquisition of increased specific infectivity during experimental evolution*’, as follows: ‘*We performed experimental evolution in a human hepatocyte cell line (Huh7.5.1. cells) as it has been previously described to be highly efficient in ZIKV production. This cell type suffers little cytopathogenic effect of ZIKV infection and is non-responsive to viral products, as it is deficient for different antiviral response sensors²⁸⁻³¹.*’ (page 5, lines 106-109)

Line 110: what is the diversity in the starting inoculum? Did you sequence your stock virus? (I see this is included later in the text but it would be worthwhile to mention the diversity in the stock inoculum.

(Reply) We have now added a sentence to present these data ahead in the first paragraph of the Result section, as follows: ‘*We sequenced the starting inoculum, and found that the major variant had on average a frequency of 99.5% (2.5% quantile: 99.2%, 97.5% quantile: 99.9%), and thus there was a standing diversity of 0.5% on average at each position.*’ (page 5, lines 111-114)

Line 115 and throughout: the use of days to represent distinct passages is confusing. Why not just use passage 1-18? The number of days is irrelevant, because as I understand it after 3 days the virus was harvested, titrated, and a new infection was started at an MOI of 0.01 PFU/cell in new cell cultures.

(Reply) To address this point, we have edited the days as passage number in all Figures and in the manuscript.

Line 121-123: specific infectivity is the ratio of infectious virus to the total amount of vRNA, which is what is shown in Fig 1A but not how it is defined in the text.

(Reply) We have now clarified the definition within the Result section. This is now added: *‘To address how passaged viral populations adapted to human host cells, we first studied their ability to initiate infection as compared to the parental virus by quantifying the specific infectivity, defined as the probability for one physical virion to initiate infection, i.e., the ratio of infectious virus relative to the total amount of vRNA (as described in Materials and Methods).’* (page 6, lines 127-131)

Fig 1A: why do lineages 2 and 3 abruptly end before lineage 1?

(Reply) We reached the plateau at these time points for the replicate, in accordance with the replicate #1, for which the experiment was of longer duration in order to validate the stability of the observed phenotype. Thus, we ended the evolution experiments for the additional round (#2 and #3) and this is now clarified in the text as follows: *‘Passaging of the virus from runs #2 and #3 was stopped when viral production reached the level at which run #1 plateaued.’* (page 6, lines 124-125)

Line 146-148: how do you explain the peaks and valleys in your specific infectivity curves? Further, it looks like the PFU:particle ratio is **1:1 at the 3 day timepoint**, is that correct, and the same as with the stock?

(Reply) These shapes resulted from the smoothing of the curves that connect the measurement of infectivity/vRNA performed at discrete time points. As also requested by Reviewer #1, we have now edited the smoothed lines in all Figures (*i.e.*, **Fig. 1a**, **Fig. 3d**, **Supplementary Fig. 1a-b** and **New Supplementary Fig. 5**). As described in the *“Material and Method”* section, the PFU:particle ratio is normalized (set to 1) and relative to the first passage of experimental evolution.

Fig 1: why only show vRNA levels here and not infectious virus?

(Reply) We selected the determination of viral RNA levels for the analysis of the kinetic of viral spread in the various cell types (**Now in Fig. 2**), because it enables: i/ to get a direct comparison between the different cell types as opposed to measurement of the level of infectious virus, which is more indirect and consequently, ii/ the determination of infectious virus would imply to use a same cell type for accurate comparison of the results yielded from different cell types and this can also impact the results.

Nonetheless, the data related to infectious virus (**Fig. 1a and Fig. 4b**) show that the levels of infectious virus *versus* vRNA provide comparable results, herein validating the quantification vRNA levels by qPCR as an accurate method in the kinetic analyses.

Line 239: it is not accurate to state that next generation sequencing cannot detect low frequency minority variants.

(Reply) We edited the text to avoid such confusion, which now reads as: *‘Sanger sequencing and next-generation deep-sequencing methods generally provide consensus sequences. The error rate of next-generation sequencing makes them ill-suited to accurately detect low frequency variants because rare variants and sequencing errors may have similar frequencies. Yet, low frequency variants can nevertheless be very important functionally.’* (page 11, lines 244-248)

Line 258: Are there any minor variants present in your stock virus?

(Reply) As above-mentioned, we now described the results of the genetic diversity in the viral inoculum used to initiate the different runs of experimental evolution, as follows: *‘We sequenced the starting inoculum, and found that the major variant had on average a*

frequency of 99.5% (2.5% quantile: 99.2%, 97.5% quantile: 99.9 and thus there was a standing diversity of 0.5% on average at each position. ’ (page 5, lines 111-114)

In addition, we now discuss the frequencies of the candidate adaptive mutations in the initial inoculum, as follows: ‘Variant C2194T was particularly rare, and variants C2341T and T7173C had unremarkable frequencies at the start of the experiment, in the middle of the distribution of variant frequencies’ (page 13, lines 298-300)

Figures: Much of the supplemental data are critical for interpretation of the results. I strongly suggest re-organizing the text and figures to logically incorporate supplemental data in the main text. In addition, I recommend revising figures to make them more intuitive and easier to read. In addition, I recommend splitting the figures into smaller multi panel figures that are inline with what is described in the text.

(Reply) Accordingly, we have reorganized the set of Figures, included additional results, and extended the set of Main Figures to 6 Figures. All **Supplementary Figures** are referenced in the main text.

Reviewers' comments:

Reviewer #1 (Remarks to the Author):

The authors satisfactorily address all my previous concerns, and the changes they have made improve the manuscript significantly and provide more detail into the mechanisms of adaptation observed in E.

There remain a few points where clarification could be helpful.

LLs. 112-114: "...the major variant had on average a frequency of 99.5%..." Here, the variant refers to the reference sequence?

LLs. 296-298: "The particularly high frequency of non-synonymous c1786T mutation, leading to reversion at position 270 of E envelope protein from V to A..." Reversion with respect to what? I think this need to be clarified.

Reviewer #2 (Remarks to the Author):

The authors made a majority of the recommended changes requested during initial peer-review of this manuscript and if the changes were not made, a sufficient explanation was provided. The changes significantly enhanced the credibility and scientific nature of the manuscript. The readers of the article can now fully understand the scientific methods used throughout this study and accurately interpret the scientific findings without bias or incomplete information.

I only have one final general comment:

Line 244-248: I still disagree with this statement that NGS cannot accurately identify low frequency variants. While it is true that deep sequencing technologies are prone to errors that can result in inaccurate variant calls—which is largely platform specific—this can be mitigated by setting stringent detection thresholds. Regardless, NGS approaches have been used to detect consensus level changes, as well as common minority SNPs that can and have been correlated to individual phenotypes, and can and have been used to inform the design of the reverse genetic and mechanistic studies for numerous viruses for variant frequencies as low as 1%.

Reviewer #3 (Remarks to the Author):

In this manuscript, authors studied the genetic evolution of ZIKA virus within hosts. Along with various experiments, the authors also used mathematical models to describe the data generated by their experiments. It is an interesting study addressing an important aspect of ZIKV infections. As my field is mathematical modeling, I primarily reviewed the modeling part of this manuscript. While it is appreciable that authors use modeling along with experimental results, I have some comments/suggestions, especially regarding the model formulation, uncertainty, and conclusions drawn based on the models.

I summarize some of my major concerns below.

1. The model seems unnecessarily complicated, causing much more uncertainty on the limited measurement available. I understand making model complicated can help include all possible mechanisms, but at the same time we introduce much more uncertainty. Therefore, it is important to justify why we need such complicated models and related terms. I suggest describing the following points.

a. I am having hard time to be convinced on why effects of both evolution and introduction of the E S455L point mutation can be modeled by the delay term in the productive replication. The experiment on intracellular ZIKV RNA (Figure 2) does not seem to support time-lag, and importantly, both magnitude and slope of the extracellular infectivity (Figure 4a) are quite different for two strains. Therefore, the better way to model and use experimental data is to have different infectivity rate (and/or fitness cost) for two strains, rather than time shift. The clarification and justification is needed in the context of experimental results.

b. In your $R(a)$ formula (Eq 4), R_0 is defined as the initial number of ZIKA RNA in the cell upon infection, but $a=0$ (age zero, i.e. cell upon infection) does not give $R(0) = R_0$. You probably need to redefine function including $R(a) = R_0$ if $a \leq \tau$.

c. Clarify what is Total cells in Eq. (5). Is it susceptible cells only or does it include infected cells as well?

d. Why is $Lipo(t)$ formula taken as complicated time-dependent expression like in Eq. (7)? How the values of two delays are obtained? Are they fitted?

e. In Eq (8), does the experiment mean time step? Clarify this as mentioning time may be important for dynamical system models. Also, what is $\gamma_{\text{simulation}}$? Clarify how you calculated $\gamma_{\text{simulation}}$ from your model, and clearly indicate what simulated values correspond to what experimental measurements (used in the fitting).

2. Clearly state that what parameters were fixed in the data fitting and what parameters were estimated. Provide both in the table so that the readers can reproduced later for further study.

3. Given limited time-points (only three time-points) of the measurement and many parameters estimated (8 parameters, supplementary Fig 7), there is a high chance that the model might have been over-fit. This may be the reason why three of the parameters do not have positive lower bound (Supplementary Figure 7), indicating the lower bound of these parameters can take unrealistic negative values. Identifiability issues need to be addressed and one way to address this is to fix some parameters (possibly from already measured experiments or from literature survey).

4. I don't think authors can explicitly conclude that the model suggests increased infectivity is due to a reduced time-lag between infection and viral replication. They have not done this study thoroughly; they have just used one model with time-lag. The possibility of τ_E having lower bound negative (Fig 7 supplementary) might indicate results in other way (increased time-lag instead of reduced time-lag). As revealed in their experiment (Fig 2), a model having different infection rates without time-lag may also explain this data.

Point-by-point response to the Reviewers

We thank the Reviewers for the positive appreciation of the revised version of our manuscript and for their helpful comments, which we have addressed by including new editions of the text.

Reviewers' comments:

Reviewer #1 (Remarks to the Author):

The authors satisfactorily address all my previous concerns, and the changes they have made improve the manuscript significantly and provide more detail into the mechanisms of adaptation observed in E.

(Reply) We thank this Reviewer for his/her general comment and specific requests for clarification.

There remain a few points where clarification could be helpful. Lls. 112-114: "...the major variant had on average a frequency of 99.5%..." Here, the variant refers to the reference sequence?

(Reply) As requested by this Reviewer, we provided clarification of the material studied (lines 111-116): 'We sequenced the starting inoculum, and found that the major variant in our viral preparation had on average a frequency of 99.5% (2.5% quantile: 99.2%, 97.5% quantile: 99.9%), and thus there was a standing diversity of 0.5% on average at each position. All positions matched the published consensus sequence except for position 1786, which had major variant T instead of C, probably from the initial amplification in Vero cells of the clinical isolate'

Lls. 296-298: "The particularly high frequency of non-synonymous c1786T mutation, leading to reversion at position 270 of E envelope protein from V to A..." Reversion with respect to what? I think this need to be clarified.

(Reply) We now made the suggested clarification (lines 298-303): '*The non-synonymous C1786T mutation was detected within the course of viral adaptation with an increasing frequency over time. This leads to a substitution at position 270 of E envelope protein from V to A. Since A270 is present in the clinical isolate database sequence (ZIKV PE243, KX197192)³⁹, V270 is likely resulting from initial viral amplification of the clinical isolate in Vero cells, and A270 is actually a reversion when passaged in Huh7.5.1 cells.*

Reviewer #2 (Remarks to the Author):

The authors made a majority of the recommended changes requested during initial peer-review of this manuscript and if the changes were not made, a sufficient explanation was provided. The changes significantly enhanced the credibility and scientific nature of the manuscript. The readers of the article can now fully understand the scientific methods used throughout this study and accurately interpret the scientific findings without bias or incomplete information.

I only have one final general comment:

Line 244-248: I still disagree with this statement that NGS cannot accurately identify low frequency variants. While it is true that deep sequencing technologies are prone to errors that can result in inaccurate variant calls—which is largely platform specific—this can be mitigated by setting stringent detection thresholds. Regardless, NGS approaches have been used to detect consensus level changes, as well as common minority SNPs that can and have been correlated to individual phenotypes, and can and have been used to inform the design of the reverse genetic and mechanistic studies for numerous viruses for variant frequencies as low as 1%.

(Reply) We agree with this point and we thank this reviewer for the request for clarification. This is now addressed as followed (lines 247-251): *‘Sanger sequencing and next-generation deep-sequencing methods generally provide consensus sequences. The error rate of next-generation sequencing can limit the detection of low frequency variants because rare variants and sequencing errors may have similar frequencies. Yet, low frequency variants can nevertheless be very important functionally.’*

Reviewer #3 (Remarks to the Author):

In this manuscript, authors studied the genetic evolution of ZIKA virus within hosts. Along with various experiments, the authors also used mathematical models to describe the data generated by their experiments. It is an interesting study addressing an important aspect of ZIKV infections. As my field is mathematical modeling, I primarily reviewed the modeling part of this manuscript. While it is appreciable that authors use modeling along with experimental results, I have some comments/suggestions, especially regarding the model formulation, uncertainty, and conclusions drawn based on the models.

I summarize some of my major concerns below.

1. The model seems unnecessarily complicated, causing much more uncertainty on the limited measurement available. I understand making model complicated can help include all possible mechanisms, but at the same time we introduce much more uncertainty. Therefore, it is important to justify why we need such complicated models and related terms. I suggest describing the following points.

a. I am having hard time to be convinced on why effects of both evolution and introduction of the E S455L point mutation can be modeled by the delay term in the productive replication. The experiment on intracellular ZIKV RNA (Figure 2) does not seem to support time-lag, and importantly, both magnitude and slope of the extracellular infectivity (Figure 4a) are quite different for two strains. Therefore, the better way to model and use experimental data is to have different infectivity rate (and/or fitness cost) for two strains, rather than time shift. The clarification and justification is needed in the context of experimental results.

(Reply) The reviewer suggests that the E S455L mutation may increase virus fitness by an increased viral infection rate. We understand the reviewer’s concern and incorporate his suggestion in our modeling. To clarify this point, we developed an alternative model with an increased virus infection rate for the E S455L mutant. The new model fits the data equally well. We show the new model in a new supplementary figure (**Supplementary Fig. 8**), and modified the text to present both models (lines 356-382): “Given that the...TLR3-induced antiviral response”. As both models, the one with E S455L shortening the delay and the other with the mutation increasing infection rate, are of the same complexity (same number of parameters), we decided to show the results of both models.

b. In your R(a) formula (Eq 4), R0 is defined as the initial number of ZIKA RNA in the cell upon infection, but a=0 (age zero, i.e. cell upon infection) does not give R(0) = R0. You probably need to redefine function including R(a) = R0 if a ≤ τ.

(Reply) This point is well taken. To remedy it, we replaced the logistic function (Eq. 4) with a piece-wise linear function of the same qualitative behavior (revised Fig. 4d). Now R0 is exactly the number of RNA molecules at infection.

c. Clarify what is Total cells in Eq. (5). Is it susceptible cells only or does it include infected cells as well?

(Reply) It includes the infected cells. To clarify this point, we explicitly wrote in Eq. 5: S(t) + I(t).

d. Why is Lipo(t) formula taken as complicated time-dependent expression like in Eq. (7)? How the values of two delays are obtained? Are they fitted?

(Reply) Following the reviewer's question, we implemented a simpler and more intuitive way of modeling the transfection and no longer use Eq. 7. This is now explained in **Appendix 1**, Mathematical model, as follows (lines 814-819): “To simulate ZIKV transfection experiment, as in Fig. 4c, we assumed that I0 cells are successfully transfected, which we estimate. Then, the number of virions produced by these cells at 24 hours post infection was simulated and used to simulate the number of infected cells at 48 hours post infection upon infection with parental or E S455L mutant virus, consistent with the experimental procedure explained for counting foci (Methods, Analysis of extracellular infectivity).”

e. In Eq (8), does the experiment mean time step? Clarify this as mentioning time may be important for dynamical system models. Also, what is y_{simulation}? Clarify how you calculated y_{simulation} from your model, and clearly indicate what simulated values correspond to what experimental measurements (used in the fitting).

(Reply) We clarify this point in the text as follows (lines 821-824):

“Parameter estimation was conducted by minimizing the weighted least-squares of the simulated values versus the experimental data

$$wSSR = \sum_{i=1}^N \sum_{j=1}^M \left[\frac{y_{simulation_{i,j}} - y_{experiment_{i,j}}}{\sigma_{i,j}} \right]^2 \quad (7)$$

where j is the experiment number and i is the data point index in time for the jth experiment as:

$$i = \left\{ \begin{array}{ll} 6, 24, 48 (h) & y_{simulation} = R_{pc}(t) \\ 48 (h) & y_{simulation} = \frac{I_{E S455L}(t)}{I_{WT}(t)} \end{array} \right\} \quad (8)$$

$$j = \left\{ \begin{array}{l} RNA_{WT}, MOI = 0.01, 0.1 \\ RNA_{E S455L}, MOI = 0.01, 0.1 \\ Infected cells ratio \end{array} \right\}.$$

2. Clearly state that what parameters were fixed in the data fitting and what parameters were estimated. Provide both in the table so that the readers can reproduced later for further study.

(Reply) We clarified this point in the **Appendix 1**, Mathematical modeling. We added the following sentence (lines 819-821): “For both the models addressing the delay or the rate

hypothesis, parameters k_{inf} , τ_P , τ_A , R_0 , R_{max} , and I_0 were estimated while the other parameters were fixed (Supplementary Fig. 7).”.

3. Given limited time-points (only three time-points) of the measurement and many parameters estimated (8 parameters, supplementary Fig 7), there is a high chance that the model might have been over-fit. This may be the reason why three of the parameters do not have positive lower bound (Supplementary Figure 7), indicating the lower bound of these parameters can take unrealistic negative values. Identifiability issues need to be addressed and one way to address this is to fix some parameters (possibly from already measured experiments or from literature survey).

(Reply) The Reviewer asked whether parameters can attain negative numbers. Clearly, the domain for all parameters are the non-negative real numbers. Hence none of the parameters can take negative values.

Further, the Reviewer suggests to fix some parameters and compute identifiability of the remaining parameters. We improved model identifiability by fixing virus production and degradation rates (k_{vp} , k_{vd}), the death rate of infected cells (k_{icd}), τ_r the time required to reach maximum replication capacity, and the intracellular ZIKV replication rate k_{rep} to the values measured/ estimated in other studies. As expected, the remaining six parameters all had positive lower bounds and except for I_0 also had upper bounds (new **Supplementary Fig. 7**). The corresponding text in Appendix 1 was amended accordingly (lines 788-809):

“where V denotes the concentration of virions. Infected cells I , have an age of infection, a , measuring how much time elapsed since infection. Their number is given by the age-structured balance equation k_{icd} :

$$\frac{\partial I}{\partial t} + \frac{\partial I}{\partial a} = -k_{icd} \cdot I(a, t) \quad (2)$$

which also accounts for cell death with rate k_{icd} which we set to 0.0144 (h^{-1})—assuming a 48 hours half-life for the infected cells (Frumence et al. 2016, Virology, <http://dx.doi.org/10.1016/j.virol.2016.03.006>). For simplicity, cell proliferation is neglected. We have the initial condition $I(a, 0) = 0$ and the boundary condition $I(0, t) = k_{inf} \cdot S \cdot V$ representing the new infections. The dynamics of the virions is obtained by integrating over infected cells of all ages:

$$\frac{dV}{dt} = k_{vp} \cdot \int_0^{72 \text{ hrs}} I(a, t) \cdot VR(a) da - k_{vd} \cdot V \quad (3)$$

where infectious virions are produced with rate k_{vp} and decay with rate k_{vd} ; these parameters were set to 41 ($\text{vir} \cdot \text{cell}^{-1} \cdot h^{-1}$) and 0.06 (h^{-1}) (Best and Perelson. 2018, Immunological reviews) ⁶².

Eq. 3 depends on the intracellular ZIKV RNA replication status $VR(a)$, which we describe by a piecewise linear function:

$$VR(a) = \begin{cases} 0 & a < \tau_x \\ k_{rep} \cdot (a - \tau_x) & \tau_x \leq a \leq \tau_x + \tau_r \\ 1 & a > \tau_x + \tau_r \end{cases} \quad (4)$$

with τ_x being the delay from infection to the onset of virus replication, and τ_r the time required to reach maximum replication capacity which we set to 20 hours as measured for Dengue virus (Talemi et al. 2021, Cell Reports). The ZIKV replication rate k_{rep} was set such that the maximum replication capacity is reached 20 hours following virus replication onset ($k_{rep} =$

0.05 h⁻¹). The delay for the parental virus τ_p and the adapted virus τ_A are estimated (**Supplementary Fig. 7**). Finally, the number of ZIKV RNA per cells was calculated as:

$$R_{pc}(t) = \frac{\int_0^{72 \text{ hrs}} I(a, t) \cdot R(a) da}{S(t) + I(t)} \quad (5)$$

where the $R(a)$ is obtained as:

$$R(a) = R_0 + R_{max} \cdot VR(a) \quad (6)$$

with R_0 being the initial number of ZIKV RNA in the cell upon infection, R_{max} the maximum number achieved by replication.”.

4. I don't think authors can explicitly conclude that the model suggests increased infectivity is due to a reduced time-lag between infection and viral replication. They have not done this study thoroughly; they have just used one model with time-lag.

The possibility of τ_E having lower bound negative (Fig 7 supplementary) might indicate results in other way (increased time-lag instead of reduced time-lag).

As revealed in their experiment (Fig 2), a model having different infection rates without time-lag may also explain this data.

(Reply) A new model considering this hypothesis is tested and incorporated in the manuscript (**Supplementary Fig. 8**). We change the main text accordingly (lines 356-382): ‘Given that the E S455L mutation causes improved viral replication both in the presence and absence of TLR3-induced signaling, we asked whether an earlier replication could provide a single mechanism explaining both observations. An earlier replication could be due to two connected processes: either an increased rate of infection (rate hypothesis), or a reduced delay between infection and viral replication (delay hypothesis). We modeled both hypotheses mathematically and compared the models and experimental data. The models share a similar backbone (**Fig. 5a**, **Supplementary Fig. 8a**) and describe the infection of susceptible cells, which, after a time delay, enter a phase of productive viral replication. Produced infectious virions can then infect other susceptible cells. Individual cells in this multiscale model will have their own time course of ZIKV replication, depending on the time of infection (**Appendix 1**, mathematical model). We determined the kinetic parameters of the models (including infection rate, delay to productive replication, and parameters defining the intracellular RNA level) by fitting the models to two sets of experimental data (**Fig. 4**, **Appendix 1**, and **Supplementary Fig. 6**). To test the delay hypothesis, the effects of both evolution and introduction of the E S455L point mutation were modeled by allowing the delay to productive replication to be different from parental and reference strains, respectively, keeping all other parameters identical. To test the rate hypothesis, we allowed the infection rate to be different from parental and reference strains (**Supplementary Fig. 8**). Both models captured the data well (**Fig. 5b-c**, **Supplementary Fig. 8b-c**). Focusing on the delay model, the parameter values were well constrained (**Supplementary Fig. 7**). Further, the delay to productive replication was shortened from around 20 hours for the controls to 10 hours or less (best fit 10 hours and upper 95% confidence bound 17 hours; **Supplementary Fig. 7**) for the mutated strains (**Fig. 5d**). Results were similar whether fitted on either data from parental versus evolved strains or data from reference versus mutated strains. Our models thus are consistent with the E S455L substitution either reducing the time-lag between infection and viral replication or increasing the infection rate (or possibly both). Both phenomena appear sufficient to cause increased viral spread, and probably the resistance to TLR3-induced antiviral response.’

REVIEWERS' COMMENTS:

Reviewer #3 (Remarks to the Author):

The authors have adequately addressed my comments. I appreciate their efforts. I recommend for publication of this manuscript in "Communications Biology".